# Conditional Advantage Estimation for Reinforcement Learning in Large Reasoning Models

**Guanxu Chen**[1,2*]    **Yafu Li**[2†]    **Yuxian Jiang**[3]    **Chen Qian**[4]    **Qihan Ren**[1]
**JingYi Yang**[3,2]    **Yu Cheng**[5]    **Dongrui Liu**[2†]    **Jing Shao**[2†]

[1] Shanghai Jiao Tong University, [2] Shanghai Artificial Intelligence Laboratory,
[3] Fudan University, [4] Renmin University of China, [5] The Chinese University of Hong Kong
*lm.cgx@sjtu.edu.cn*        *yafuly@gmail.com*        *{liudongrui, shaojing}@pjlab.org.cn*

## Abstract

Reinforcement Learning with Verifiable Rewards (RLVR) for large language models (LLMs) has achieved remarkable progress in enhancing LLMs' reasoning capabilities on tasks with clear correctness criteria, such as mathematical reasoning tasks. Several training metrics, such as entropy or response length, have been observed to correlate with different reasoning behaviors in reinforcement learning. Prior approaches incorporate such priors through reward or advantage shaping, which often relies on hand-crafted penalties and preferences (e.g., higher-is-better or lower-is-better). However, without careful hyper-parameter tuning, these directional priors can be overly biased and may lead to failure. To this end, we introduce **Conditional advANtage estimatiON** (CANON), amplifying the impact of the target metric without presuming its direction. Specifically, CANON regroups the sampled responses into two groups based on the higher or lower value of a target metric, measures which metric trend contributes to better performance through inter-group comparison, and identifies the better response within the same group. In summary, CANON based on entropy consistently outperforms prior methods across three LLMs on both math reasoning and high-complexity logic tasks. When applied to response length, CANON further improves token efficiency, yielding a more favorable Pareto frontier in the performance–cost trade-off.

## 1 Introduction

Recently, Large Reasoning Models (LRMs) such as Gemini 2.5 Pro (Comanici et al., 2025), DeepSeek-R1 (Guo et al., 2025), and OpenAI-o1 (Jaech et al., 2024), continue to push the boundaries of performance on reasoning tasks. A key technique driving this success is Reinforcement Learning with Verifiable Rewards (RLVR), which enables models to refine answers through multi-step reflection. Algorithms designed for RLVR, most prominently GRPO (Shao et al., 2024) and its variants (e.g., DR.GRPO, Liu et al. (2025a)), have become central to achieving superior performance.

In previous works, some training metrics are observed to be closely correlated with model behavior, which can guide the training process and improve LLMs' performance (Hassid et al., 2025; Gandhi et al., 2025; Wang et al., 2025). To incorporate such a human prior, some methods integrate these metrics through reward shaping (Arora & Zanette, 2025; Luo et al., 2025) and advantage shaping (Chen et al., 2025b; Cheng et al., 2025) to guide the model's reasoning behavior. For example, an over-length penalty is used to boost reasoning efficiency, and the entropy signal is leveraged to maintain exploration for better performance.

However, these methods usually introduce human priors by adding penalty and reward terms, which hold handcrafted priors that specific metrics are either to be higher-is-better or to be lower-is-better. Without careful hyper-parameter selection, these priors can be overly biased and drive specific metrics up or down directly, thus failing to enhance performance robustly. Simple handcrafted priors towards one specific direction are hard to work in different scenarios. For instance, higher-entropy responses tend to be exploratory and may correctly answer complex questions, whereas lower-entropy responses

---

*Our code is available at ⊙CANON.
†Corresponding Author.

exhibit higher certainty and achieve greater accuracy on most questions within their capability (Cheng et al., 2025; Prabhudesai et al., 2025; Wang et al., 2025). Therefore, we aim to amplify the impact of specific metric changes without presupposing preferences, naturally identifying inherent tendencies in model rollouts that can be leveraged to facilitate learning of beneficial behaviors, such as enhancing exploration or improving reasoning efficiency.

To this end, we regroup the sampled responses into two groups based on the higher or lower values of a given metric during the process of RLVR training. Specifically, we can sort the sampled responses according to the value and split them into two groups. Based on this, we propose *Conditional advANtage estimatiON* (`CANON`), which computes the inter-group advantage by comparing a response with the group that it does not belong to, and gets the intra-group advantage across its own group conversely. The inter-group advantage reveals which trend of metrics leads to higher accuracy. Meanwhile, the intra-group advantage identifies better responses within the same group.

Taking the metric of entropy as an example, if groups with lower entropy (i.e., higher certainty) yield higher average rewards, the inter-group advantage tends to select correct responses with low entropy, efficiently exploiting existing features to boost performance. In contrast, correct rollouts with higher entropy receive more advantages in the intra-group comparison because the average reward of their group is lower, thereby encouraging truly effective exploration. We theoretically prove that when the two groups have equal size, the inter-group advantage amplifies the impact of the grouping metric on the advantage computation. In this setting, DR.GRPO can be formulated as a uniform weighting of these two advantages, which is a special case of `CANON`.

We consider the metrics of generation entropy and response length, evaluating the effectiveness of `CANON` on three open-weight LLMs across six math reasoning benchmarks and three challenging logic reasoning tasks. Empirical results show that emphasizing the inter-group advantage based on entropy yields a **1.9**-point accuracy gain on math tasks. In contrast, for high-complexity reasoning problems, the intra-group advantage proves crucial, achieving a **5.2**-point improvement on the most challenging subset. Through scheduling of these advantages, `CANON` further achieves a superior and comprehensive performance across three models and two tasks. Furthermore, `CANON` based on response length substantially enhances reasoning efficiency, establishing a new Pareto frontier in the performance–efficiency trade-off. In low-token-budget scenarios for math tasks, it achieves **2.63×** higher performance and reduces token consumption by **45.5%** at the same performance level.

## 2    RELATED WORK

**Advantage Estimations in Reinforcement Learning.** In PPO, the advantage estimation is provided by Generalized Advantage Estimation (GAE, Schulman et al. (2015)).To avoid the computational cost of the critic model, several methods, such as ReMax (Li et al., 2023), RLOO (Ahmadian et al., 2024), GRPO Shao et al. (2024), and REINFORCE++ (Hu, 2025), utilize alternative techniques like baseline reward and group-relative rewards for advantage estimation. ReMax compares the rewards with the baseline reward from the greedy decoding response. REINFORCE++ estimates the advantage by the normalization operation across the global batch for all queries. RLOO and GRPO estimate the advantage in a group relative manner. RLOO computes the average rewards of all other solutions in the group as the baseline reward, and GRPO utilizes the normalized rewards among the sampled solutions as the advantage estimation. Compared to GRPO, our method splits sampled responses into two groups based on specific conditions and selects the appropriate condition through inter- and intra-group comparisons, thereby efficiently optimizing key patterns that boost task performance.

**Reinforcement Learning with Verifiable Rewards.** RLVR leverages the existing RLHF objective (Schulman et al., 2017) but replaces the reward model with a verification function, which is available in domains with verifiable answers, such as mathematics reasoning tasks (Guo et al., 2025; Lambert et al., 2024). Yu et al. (2025); Liu et al. (2025b); Chen et al. (2025a) consider the importance sampling techniques and contribute novel training paradigms and optimization objectives for better and more stable reasoning capabilities. Due to the sparse rewards during training, past methods utilize not only accuracy-based rewards but also explicitly integrate additional signals through reward shaping (Arora & Zanette, 2025; Luo et al., 2025) and advantage shaping (Chen et al., 2025b; Cheng et al., 2025) to guide the model's reasoning and reflection. Arora & Zanette (2025) and Luo et al. (2025) utilize an over-length penalty to boost reasoning efficiency. Chen et al. (2025b) and (Cheng et al., 2025) consider the entropy as a measure of exploration and reshape the advantage computation. Gandhi et al. (2025) also observes four key cognitive behaviors of initial reasoning behaviors and strengthens

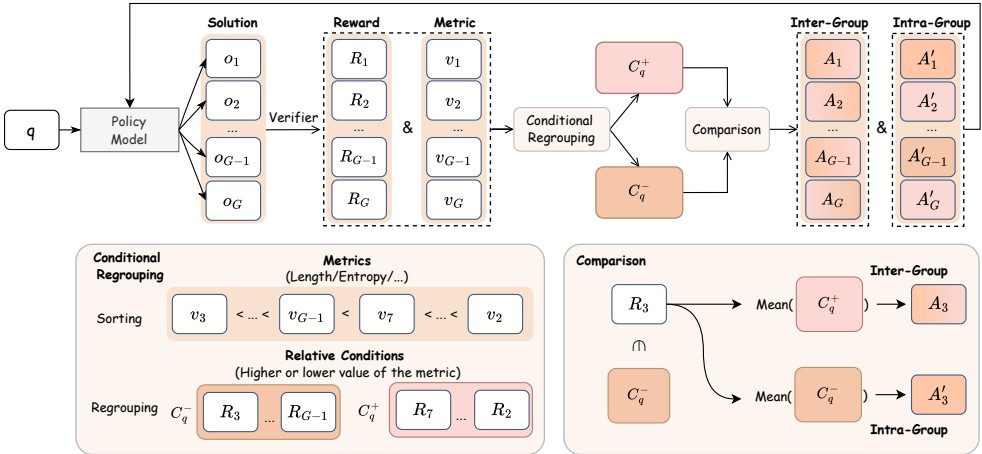

Figure 1: Overview of CANON. CANON regroups all the sampled responses based on the value of a specific metric, and computes the advantages through inter-group and intra-group comparison.

the capacity for self-improvement. However, these methods usually introduce human priors by adding penalty and reward terms, which hold handcrafted priors that can be overly biased and may fail to enhance performance without careful hyper-parameter selection. Our work amplifies the impact of specific metric changes without presupposing preferences, leveraging them to facilitate learning of beneficial behaviors.

## 3    PRELIMINARIES

Proximal Policy Optimization (PPO, Schulman et al. (2017)) is a widely used method for policy optimization of LLMs. PPO utilizes the clip mechanism to update policy stably. PPO maximizes the following optimization objectives.

$$\mathcal{J}_{\text{PPO}}(\theta) = \mathbb{E}_{q\sim\mathcal{D},o\sim\pi_{\theta_{\text{old}}}(\cdot|q)} \left[ \frac{1}{|o|} \sum_{t=1}^{|o|} \min\left( r_{o,t}(\theta)\hat{A}_t,\ \text{clip}_{1-\varepsilon}^{1+\varepsilon}(r_{o_i,t}(\theta))\hat{A}_t \right) \right], \quad (1)$$

where $\pi_{\theta_{\text{old}}}$ and $\pi_\theta$ are used to denote the policy model before and after the update. $q$ is a query sampled from the data distribution $\mathcal{D}$, and the output $o$ is generated by $\pi_{\theta_{\text{old}}}$. The clipping function with clip ratio $\varepsilon$ is computed as $\text{clip}_a^b(x) = \max(\min(x,a),b)$ and the importance sampling ratio at time step $t$ is defined as $r_{o,t}(\theta) = \frac{\pi_\theta(o_t|q,o_{<t})}{\pi_{\theta_{\text{old}}}(o_t|q,o_{<t})}$.

To avoid the computational cost of the critic model, GRPO (Shao et al., 2024) estimates the advantage in a group relative manner. They sample $G$ different solutions for the current query $q$ as the group $G_q := \{o|o \sim \pi_{\theta_{\text{old}}}(.|q)\}$, and calculate the normalized rewards as advantages within the group $G_q$.

$$\hat{A}_{q,o,t}^{\text{GRPO}} = \frac{R_o - \text{mean}(\{R_{o'}|o' \in G_q\})}{\text{std}(\{R_{o'}|o' \in G_q\})}. \quad (2)$$

Due to the success of DeepSeek-R1, several studies have proposed improvements based on GRPO. DR.GRPO (Liu et al., 2025a) uses the GRPO advantages without standard deviation normalization and develops a token-level loss without length bias.

## 4    CONDITIONAL ADVANTAGE ESTIMATION

Group-based advantage estimation methods, such as GRPO, typically use the average reward of all sampled responses within the group as a baseline reward. This may fail to provide a clear feedback signal for policy optimization due to the ambiguity of the comparison target. We propose CANON, which performs conditional regrouping by splitting all sampled responses into two groups based on the value of a specific metric. Leveraging these two groups, inter-group advantage identifies the metric trend that yields higher accuracy through cross-group comparison, while intra-group advantage

selects superior responses within the same trend and prioritizes correct answers from the group with a lower average reward.

## 4.1 CONDITIONAL REGROUPING

To explicitly introduce a comparison target, we regroup all the sampled responses based on specific conditions. Given any condition $c$, we denote the set of all outputs for the current query $q$ that satisfy this condition in the sampled group $G_q$ as $C_q^+ := \{o | o \text{ satisfy } c, o \in G_q\}$. The set of outputs that do not satisfy the condition can be denoted by $C_q^- = G_q \setminus C_q$. In this work, we focus on studying the relative conditions given by the training metrics, such as the entropy and length of the sampled responses. Specifically, we divide the responses into two non-overlapping groups based on the value of the metrics, as shown in Figure 1.

## 4.2 ADVANTAGE ESTIMATION BASED ON REGROUPING.

Given two groups, we can compute the inter-group advantage through comparison between different groups.

$$\hat{A}_{q,o,t}^{\text{inter}} = \begin{cases} R_o - \text{mean}(\{R_{o'} | o' \in G_q^+\}), \text{if } o \in G_q^- \\ \\ R_o - \text{mean}(\{R_{o'} | o' \in G_q^-\}), \text{if } o \in G_q^+ \end{cases} . \tag{3}$$

Meanwhile, we also compute the intra-group advantage by comparing each response with the mean reward of its own group.

$$\hat{A}_{q,o,t}^{\text{intra}} = \begin{cases} R_o - \text{mean}(\{R_{o'} | o' \in G_q^+\}), \text{if } o \in G_q^+ \\ \\ R_o - \text{mean}(\{R_{o'} | o' \in G_q^-\}), \text{if } o \in G_q^- \end{cases} . \tag{4}$$

Although this may appear similar to the estimation of DR.GRPO within a smaller scope, due to the differing average advantages between groups, the intra-group advantage prioritizes correct responses from the group with a lower average reward ($1 - \text{mean}(\{R_{o'} | o' \in G_q^+\} > 1 - \text{mean}(\{R_{o'} | o' \in G_q^-\}$ when $\text{mean}(\{R_{o'} | o' \in G_q^+\} < \text{mean}(\{R_{o'} | o' \in G_q^-\}$ ). We can further combine the above two advantages into a unified formulation.

$$\hat{A}_{q,o,t}^{\text{CANON}} = \mu \hat{A}_{q,o,t}^{\text{inter}} + (1-\mu)\hat{A}_{q,o,t}^{\text{intra}}, \tag{5}$$

where $\mu$ controls the balance between the inter-group and intra-group advantage. Figure 1 demonstrates a concise case of the computation of CANON.

To ensure that the advantages introduced by conditional regrouping provide a clearer contrastive signal, we theoretically analyze the situations under which inter-group advantage, compared to DR.GRPO, yields a stronger advantage signal in response to reward gaps under specific conditions.

**Theorem 1** (Situations with clearer advantage signal (proved in Appendix E)). *Suppose that condition $c$ is based on numerical comparisons and can be derived through sorting of metrics. Further assume that the sampled response $o$ to query $q$ satisfy condition $c$ with probability $p \in (0,1)$, and $\mathbf{E}_{o \text{ satisfy } c}[R_o] \neq \mathbf{E}_{o \text{ not satisfy } c}[R_o]$. Then, we have:*

$$\frac{|\hat{A}_{q,o,t}^{inter}|}{|\hat{A}_{q,o,t}^{DR.GRPO}|} > 1, \text{ only when } |C_q^+| = |C_q^-| \text{ if } |C_q^+| \text{ is a constant.} \tag{6}$$

Based on Theorem 1, we divide the responses into two equally sized groups. In this way, DR.GRPO can be expressed as a special case of this unified form when $\mu = 0.5$.

$$\hat{A}_{q,o,t}^{\text{DR.GRPO}} = R_o - \text{mean}(\{R_{o'} | o' \in G_q\}) = \frac{1}{2}\hat{A}_{q,o,t}^{\text{inter}} + \frac{1}{2}\hat{A}_{q,o,t}^{\text{intra}}. \tag{7}$$

Moreover, rather than a direct numerical amplification, CANON amplifies only the advantage attributable to the metric used for grouping, without amplifying the influence of other factors.

Table 1: Overall performance based on **Qwen2.5-Math-7B**. We compare with the following baselines: (1) Qwen2.5-Math-7B-Instruct (Qwen-Instruct), (2) prior advantage estimation methods. All models are evaluated under a unified setting. **Bold** and underline indicate the best and second-best results, respectively.

| Model | Math Reasoning | | | | | | | | High Complexity Reasoning | | | | |
|---|---|---|---|---|---|---|---|---|---|---|---|---|---|
| | AIME 24 | AIME 25 | Olympiad | AMC | MATH-500 | GSM8k | Tokens | Acc | Mid | Large | XLarge | Tokens | Acc |
| Base | 16.0 | 8.0 | 26.4 | 41.6 | 61.2 | 61.6 | 2046 | 35.8 | 0.0 | 0.5 | 0.1 | 3303 | 0.2 |
| Instruct | 10.7 | 9.7 | 39.7 | 49.3 | 82.2 | 94.8 | 1077 | 47.7 | 11.6 | 6.2 | 3.5 | 2647 | 7.1 |
| *Previous Advantage Estimation* | | | | | | | | | | | | | |
| ReMax | 23.3 | 18.0 | 48.1 | 62.8 | 83.4 | 90.3 | 2418 | 54.3 | 37.2 | 21.0 | 9.7 | 6246 | 22.6 |
| R++ | 20.3 | 19.7 | 45.8 | 58.3 | 82.6 | 90.0 | 4107 | 52.8 | 33.8 | 11.9 | 3.3 | 9923 | 16.3 |
| RLOO | 25.0 | 18.7 | 51.3 | **64.3** | 84.0 | 91.0 | 2537 | 55.7 | 33.9 | 14.4 | 5.8 | 10610 | 18.0 |
| GRPO | 22.3 | 18.3 | 47.3 | 60.6 | 83.8 | 90.8 | 3730 | 53.8 | 31.5 | 14.9 | 5.2 | 9406 | 17.2 |
| DR.GRPO ($\mu = 0.5$) | 27.7 | 20.3 | 48.4 | 63.4 | 83.2 | 91.1 | 1522 | 55.7 | 39.2 | 24.4 | 15.1 | 4896 | 26.2 |
| *Entropy-related Baselines* | | | | | | | | | | | | | |
| Entropy Adv | 26.7 | 16.7 | 50.8 | 65.3 | **87.6** | 90.8 | 2389 | 56.3 | 30.8 | 17.1 | 7.5 | 8207 | 18.5 |
| Clip-Cov | 26.3 | **21.0** | 49.0 | 63.5 | 84.8 | 92.1 | 1344 | 56.1 | 39.2 | 25.6 | 14.7 | 4045 | 26.5 |
| *Our Methods (Conditional Groups based on Length)* | | | | | | | | | | | | | |
| CANON-Intra | 21.7 | 19.0 | 49.9 | 63.0 | 86.2 | **92.2** | 2176 | 55.3 | 41.8 | 25.6 | 14.7 | 4364 | 27.4 |
| CANON-Inter | 27.3 | 19.3 | 47.6 | 64.2 | 82.6 | 91.0 | **1008** | 55.3 | **42.7** | **28.6** | 17.1 | 3652 | **29.5** |
| *Our Methods (Conditional Groups based on Entropy)* | | | | | | | | | | | | | |
| CANON-Intra | 25.0 | 16.0 | 48.9 | 62.7 | 84.4 | 91.1 | 2959 | 54.7 | 39.1 | 27.8 | **20.3** | **3101** | 29.1 |
| CANON-Inter | **32.7** | 18.7 | **51.7** | 64.2 | 87.0 | 91.1 | 1466 | **57.6** | 36.3 | 25.8 | 14.9 | 4415 | 25.7 |
| CANON-Dynamic | 30.0 | 17.7 | 50.7 | 63.3 | 86.6 | 91.8 | 1452 | 56.7 | 40.4 | 30.5 | 16.6 | 3535 | 29.2 |

**Theorem 2** (Selective amplification based on specific metrics (proved in Appendix E)). *Consider independent conditions $c_1$ and $c_2$, and their corresponding sets $C_1$ and $C_2$ (i.e., $P(o \in C_1 \cap C_2|q,\theta) = P(o \in C_1|q,\theta)P(o \in C_2|q,\theta)$). When we fix the condition $c_1$, then for any value of the performance gap $a_{2+}$, $a_{2-}$ and the probability $P(o \in C_2|q,\theta)$ that induced by whether $c_2$ is satisfied, we have*

$$\frac{|\hat{A}_{q,o,t}^{inter\ based\ on\ c_1}|}{|\hat{A}_{q,o,t}^{DR.GRPO}|} \text{ is a constant.} \tag{8}$$

*which says* CANON *based on the condition $c_1$ will not amplify the influence of another independent condition $c_2$.*

Therefore, CANON, when grouped by a specific metric, amplifies the influence of that metric during training, yet it does not predefine a preference for the magnitude of the metric. This design allows it to incorporate human priors while mitigating bias, which fully aligns with our original motivation.

### 4.3 ALIGNING WITH TRAINING TARGET THROUGH WEIGHTED ADVANTAGE

According to Section 4.2, the selection between different trends of metrics only takes place in the inter-group advantage. By weighting different conditions within the inter-group advantage calculation, this enables fine-grained control over the trend of metrics with only tiny differences compared to DR.GRPO. For instance, by slightly reducing the weight of longer responses, CANON can accomplish reasoning of high token efficiency through the RL process. Specifically, the inter-group advantage in the Eq. 5 should be replaced with $\hat{A}_{q,o,t,\alpha}^{\text{inter}}$ where $\alpha$ is the weight of a specific group, and $\hat{A}_{q,o,t,\alpha}^{\text{inter}}$ is defined as:

$$\hat{A}_{q,o,t,\alpha}^{\text{inter}} = \begin{cases} R_o - \alpha * \text{mean}(\{R_{o'}|o' \in G_q^+\}), \text{if } o \in G_q^- \\ \alpha * R_o - \text{mean}(\{R_{o'}|o' \in G_q^-\}), \text{if } o \in G_q^+ \end{cases}. \tag{9}$$

For example, setting $\alpha$ as 0.9 can achieve substantial length reduction with little performance drop, where $C_q^+$ is considered the group with longer responses.

## 5 EXPERIMENTS

The empirical evaluation of CANON consists of three parts. Firstly, we demonstrate the effect of intra-group and inter-group advantages, respectively, across six math reasoning benchmarks and one

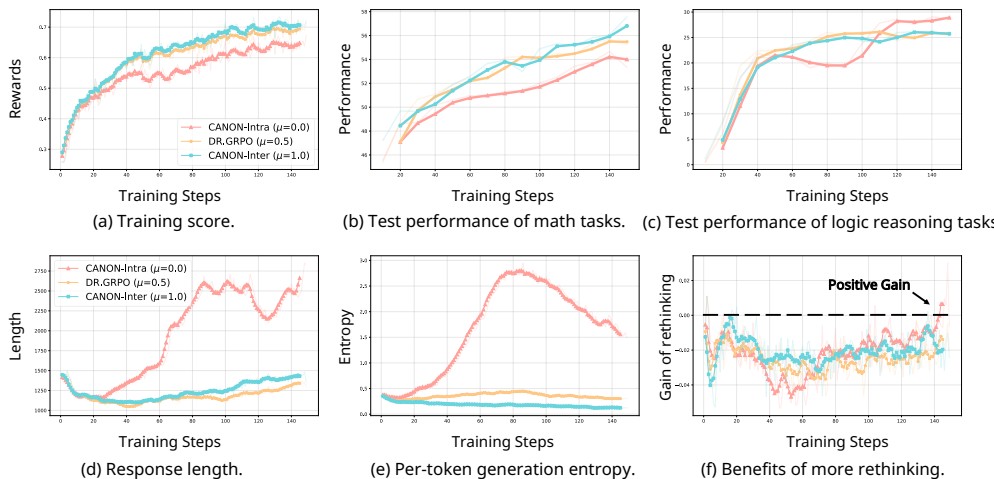

Figure 2: The training dynamics and average test performance of `CANON-Inter`, DR.GRPO, and `CANON-Intra`.

high-complexity logic reasoning benchmark. In the second part, we perform several scheduling tricks to get the frontier in both tasks. At last, by weighting the longer responses with $\alpha < 1$, we achieve efficient reasoning that reaches a better Pareto frontier.

## 5.1 PERFORMANCE OF INTRA-GROUP AND INTER-GROUP ADVANTAGES.

**Training Setup.** We select the response length and the per-token generation entropy, respectively, to regroup the sampled solutions. We use a subset with 45k prompts from OpenR1-Math-220k (Hugging Face, 2025) that is filtered and constructed by Yan et al. (2025). Following DR.GRPO (Liu et al., 2025a) and DAPO (Yu et al., 2025), we correct the response-level length bias and utilize the clip-higher strategy ($\epsilon_{high} = 0.28$) for all experiments. We also remove both the KL loss and the entropy loss. We sample 16 responses per prompt and use temperature=1.0 for rollout generation. Our rollout batch size is 512, and the train batch size is 32. The responses to the same prompt are separated into two evenly sized groups by sorting ordinal variables. We conduct the main experiments on Qwen2.5-Math-7B (Yang et al., 2024) following Zeng et al. (2025); Liu et al. (2025a); Yan et al. (2025). We expand Qwen2.5-Math-7B's context limit from 4096 to 16384 by changing the rope theta from 10000 to 40000[1]. We set the maximum answer length to 8192 and the learning rate is set to 1e-6. We use *Math-Verify* to give the 0-1 score for both training reward and evaluation accuracy.

**Evaluation Setup.** We evaluate the math reasoning capabilities on six commonly used benchmarks, such as MATH-500 (Hendrycks et al., 2021), GSM8K (Cobbe et al., 2021), AMC (Li et al., 2024), OlympiadBench (He et al., 2024), and AIME 24/25. Due to the tiny size of AIME 24/25 and AMC, we report *Avg@10* as the test accuracy. For the other benchmarks, we compute the *Pass@1* as the test performance. We calculate the average performance and token cost across all benchmarks. All models are evaluated under the same setting with a temperature of 0.6. The values in Table 1 are the percentage accuracy of the models evaluated. We also select three high-complexity subsets of ZebraLogic (Lin et al., 2025) with their solution space sizes greater than $10^3$ (Mid), $10^6$ (Large), and $10^9$ (XLarge), respectively. In this experiment, we record six metrics, including training reward, generation entropy, response length, the test performance of math tasks and logic reasoning task, and the marginal improvement gained from reflection.

**Baselines.** In this subsection, we fix $\alpha = 1.0$ in Eq. 9 and present the results of $\mu = 0.0$ (`CANON-Intra`) and $\mu = 1.0$ (`CANON-Inter`) in Eq. 5. A more detailed scheduling on $\mu$ will be conducted in Section 5.2, and the adjustment of $\alpha$ will be covered in Section 5.3. We compare `CANON` with two types of baselines: (1) **Qwen2.5-Math-7B-Instruct** (Instruct, Yang et al. (2024)), (2) **previous advantage estimation methods**, such as ReMax, REINFORCE++ (R++), RLOO, GRPO, and DR.GRPO, and (3) **entropy-related baselines**, such as Entropy Adv (Cheng et al., 2025) and Clip-Cov (Cui et al., 2025).

---

[1]The original context limit leads to unacceptable length clipping ratio. Please see Figure 7 in Appendix C.3.

Table 2: Overall performance of CANON-Dynamic across three different models and two tasks. All models are evaluated under a unified setting. **Bold** and underline indicate the best and second-best results, respectively.

| Model | Math Reasoning | | | | | | | | High Complexity Reasoning | | | | |
|---|---|---|---|---|---|---|---|---|---|---|---|---|---|
| | AIME 24 | AIME 25 | Olympiad | AMC | MATH-500 | GSM8k | Tokens | Acc | Mid | Large | XLarge | Tokens | Acc |
| *Qwen2.5-Math-7B* | | | | | | | | | | | | | |
| DR.GRPO ($\mu = 0.5$) | 27.7 | **20.3** | 48.4 | 63.4 | 83.2 | 91.1 | 1522 | 55.7 | 39.2 | 24.4 | 15.1 | 4896 | 26.2 |
| *Cosin-First-Inter-Later-Intra* | **30.0** | 17.7 | 50.7 | 63.3 | **86.6** | 91.8 | 1452 | 56.7 | 40.4 | 30.5 | 16.6 | 3535 | **29.2** |
| *First-Inter-Later-Intra* | 28.0 | **20.3** | 52.4 | 64.6 | 84.2 | 92.6 | **1328** | **57.0** | **41.7** | 26.6 | 16.5 | 3862 | 28.3 |
| *Qwen2.5-Math-1.5B* | | | | | | | | | | | | | |
| DR.GRPO ($\mu = 0.5$) | 13.3 | 11.0 | **43.9** | 48.8 | 77.0 | 84.3 | 2381 | 46.4 | 23.7 | 9.7 | 5.0 | 9215 | 12.8 |
| *Cosin-First-Inter-Later-Intra* | **17.3** | 13.7 | 40.6 | 50.0 | 76.0 | 83.9 | 2357 | **46.9** | 19.2 | 8.9 | 4.2 | 10382 | 10.8 |
| *First-Inter-Later-Intra* | 16.0 | 10.0 | 42.4 | 50.2 | **78.6** | 83.3 | 2479 | 46.8 | **27.0** | 16.3 | 7.9 | 7070 | **17.0** |
| *Llama3.1-8B* | | | | | | | | | | | | | |
| DR.GRPO ($\mu = 0.5$) | 1.3 | **0.3** | 8.3 | 11.3 | 32.0 | 78.9 | 9476 | 22.0 | 21.1 | 13.8 | 9.7 | 5864 | 14.9 |
| *Cosin-First-Inter-Later-Intra* | 0.7 | 0.0 | 7.1 | **12.4** | **33.8** | **81.4** | 2354 | **22.6** | 26.0 | 18.4 | 12.3 | 1685 | **18.9** |
| *First-Inter-Later-Intra* | **2.0** | 0.0 | **8.7** | 9.9 | 31.8 | 80.1 | 3488 | 22.1 | 25.1 | 17.5 | 10.6 | 5892 | 17.7 |

**Inter-group advantage achieves higher accuracy and lower length in math tasks.** The experimental results are shown in Table 1. CANON-Inter based on *Entropy* achieves an average performance of 57.6 among six math benchmarks, which is 1.9 points higher than the DR.GRPO (55.7). Specifically, CANON-Inter based on *Entropy* has the best performance on four of the six benchmarks, and is highly competitive with the top-performing models on the rest. In AIME24, the model's performance is 5.0 points higher than the DR.GRPO's. Meanwhile, CANON-Inter based on *Length* reduces the token cost by 33.8% compared with DR.GRPO, while maintaining nearly unchanged performance (55.7 vs. 55.3).

**The benefit of intra-group advantage grows as the logic reasoning task's complexity increases.** Table 1 demonstrates that CANON-Intra based on *Entropy* achieves higher performance of 2.9 points and 36.6% shorter length compared with DR.GRPO. Its performance edge over DR.GRPO increases (from -0.1 to 3.4 and then 5.2) when the complexity becomes higher. The results of CANON-Intra based on *Length* shows another trend, whose inter-group advantage makes the best performance in this task.

**Training dynamics reflect different roles of CANON-Intra and CANON-Inter.** To be specific, we record training curves under the setting of CANON based on *Entropy*. The training dynamic shown in Figure 2 indicates that both the training reward and the test performance of the math tasks increase rapidly when only CANON-Inter is utilized ($\mu = 1.0$). Its generation entropy stably decreases, and the response length changes smoothly. When using only CANON-Intra ($\mu = 0.0$), the responses show a greater tendency for exploration. We divide the responses into two groups by counting reflection patterns and calculate the gap in average reward between the group with more and fewer reflections (Figure 2f). Figure 2 demonstrates that the trend of high-complexity reasoning performance is highly consistent with the curve of reflection gains. In the later stages of training (after approximately 90 steps), the reflection gain curve of intra-group advantage increases and finally crosses the zero point. At the same time, its performance experiences rapid growth, significantly outperforming the other two advantages.

## 5.2 BALANCING PERFORMANCE VIA ADVANTAGE SCHEDULING

As shown in Table 1 and Figure 2, CANON-Inter and CANON-Intra outperform DR.GRPO on the math reasoning task and the complex logic reasoning task, respectively, but neither can achieve the best performance on both simultaneously. To this end, we schedule the CANON-Inter and CANON-Intra by leveraging accuracy and the training steps to achieve a better balance between the two scenarios.

**Setup.** We conduct experiments across six math benchmarks and three complex logic reasoning tasks on Qwen2.5-Math-7B (Yang et al., 2024), Llama3.1-8B (Dubey et al., 2024), and Qwen2.5-Math-1.5B (Yang et al., 2024). For the two Qwen series models, we use the dataset introduced in Section 5.1. Due to the weak capability of Llama3.1-8B, we collect a simpler dataset with 35k samples from four open-source datasets and follow the other training setups described in Section 5.1. Please see the

details of this newly constructed dataset in Appendix C.5. We draw a radar chart with the average performance of the two scenarios for visualization, and the results for `CANON` with scheduling are denoted as `CANON-Dynamic`.

**Scheduling strategies.** All of the strategies are based on the coefficient $\mu$ in the Eq. 5, which balances the `CANON-Inter` and `CANON-Intra`. We try four scheduling strategies utilizing the training accuracy and training steps, respectively: (1) *First-Inter-Later-Intra*. We set the value of $\mu$ to $1 - \Lambda$, where $\Lambda$ denotes the mean accuracy of current whole batch; (2) *First-Intra-Later-Inter*. We set the value of $\mu$ to $\Lambda$. (3) *Cosin-First-Inter-Later-Intra*. We schedule the value of $\mu$ from high to low using a cosine annealing function with restarts and warm-up. (4) *Cosin-First-Intra-Later-Inter*. We schedule the value of $\mu$ from low to high using a cosine annealing function with restarts and warm-up. Please see Appendix C.6 for more details. The shown results of `CANON-Dynamic` are derived from one of the tried scheduling strategies that achieve strong performance in both scenarios.

***First-Inter-Later-Intra* consistently performs better than DR.GRPO across three models and two tasks.** As shown in Table 2, all three models demonstrate the same trend that performs better than the baseline by first applying Inter-group advantage and then using Intra-group advantage. Qwen2.5-1.5B performs particularly well under accuracy-based scheduling, possibly because its training accuracy range (0–0.6) aligns well with its learning progress. In contrast, the other two models may achieve higher final accuracies, which—under the same scheduling scheme—trigger excessive exploration and consequently lead to suboptimal final performance. We utilize fixed min/max values of $\mu$ by applying cosine annealing based on training steps, achieving higher performance.

Moreover, different models may have different numbers of parameters and different levels of capability. A specifically designed strategy is acceptable for better performance in practice. In this way, we select strategy *Cosin-First-Inter-Later-Intra* for Qwen2.5-Math-7B and Llama3.1-8B, and strategy *First-Inter-Later-Intra* for Qwen2.5-Math-1.5B to draw Figure 3. As shown in Figure 3, `CANON-Dynamic` outperforms DR.GRPO across all models and tasks, achieving a superior and more comprehensive performance. Although its math performance on Qwen2.5-Math-7B lags slightly behind `CANON-Inter`, it still makes a better performance than DR.GRPO. The radar chart illustrates the trade-off between two types of tasks faced by `CANON-Inter` and `CANON-Intra` between two types of tasks, as well as the balanced but mediocre performance of DR.GRPO.

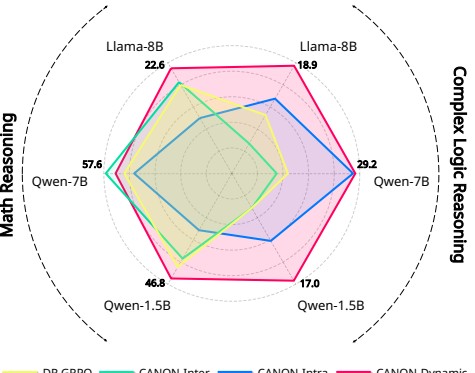

Figure 3: Evaluation for three LLMs across two types of reasoning tasks. We apply a model-specific schedule for a given model that consistently yields leading results across both mathematical and logical reasoning tasks

## 5.3 WEIGHTED CONDITIONS FOR EFFICIENT REASONING.

**Training Setup.** In this subsection, we utilize `CANON` based on response length with $\mu = 0.5$ in the Eq. 5 and tune the $\alpha$ in the Eq. 9, where $C_q^+$ is considered the group with longer responses. A larger $\alpha$ means less compression of length. We follow the training setups described in Section 5.1 and reduce the maximum response length to 3072 for better efficiency. To be specific, we use `CANON-Eff` to denote the results of `CANON` with weighted conditions of length.

**Evaluation Setup under different token budgets and varying hyperparameter settings of different methods.** To systematically assess LRMs' reasoning efficiency (Qu et al., 2025), we introduce two types of curves: **budget-performance curves for each LRM** and **cost-performance curves of different coefficients for all compared baselines**. Specifically, we set a maximum budget for each benchmark based on its difficulty and the average unconstrained output length of LRMs (Appendix C.2), then slice the same response at various budget ratios to draw the budget-performance curves. Moreover, we tune the length-controlling coefficients of each baseline to draw the cost-performance curves, recording their average performance and token cost to enable a comprehensive and fair comparison. Please see the subsection on Pareto frontier for the specific hyperparameters. In every comparison, the closer to the upper-left corner, the better (which represents high accuracy and high efficiency at the same time).

Table 3: The comparison between different methods towards efficient reasoning. Bold and underline indicate the best and second-best results, respectively. The detailed performance is from the top-performing models for each method, specifically $\alpha$=0.96 for CANON-Eff. We include CANON-Eff with $\alpha = 0.88$, which has comparable performance with the baseline Length Reward (*).

| | AIME 24 | | AIME 25 | | Olympiad | | AMC | | MATH-500 | | GSM8k | | Overall | |
|---|---|---|---|---|---|---|---|---|---|---|---|---|---|---|
| | Acc | Tokens | Acc | Tokens | Acc | Tokens | Acc | Tokens | Acc | Tokens | Acc | Tokens | Acc | Tokens |
| DR.GRPO | 29.0 | 1640 | 19.0 | 1586 | 49.0 | 1172 | 64.6 | 1214 | 85.8 | 728 | 91.9 | 349 | 56.6 | 1115 |
| Clip Length | 28.0 | 1177 | 18.3 | 1177 | 47.3 | 915 | 63.1 | 956 | 84.8 | 612 | 92.9 | 291 | 55.7 | 855 |
| Length Reward$_+$ | 31.7 | 1190 | 18.0 | 1208 | 46.7 | 864 | 61.8 | 937 | 84.6 | 546 | 91.9 | 255 | 56.2 | 869 |
| Length Reward$_*$ | 27.3 | 1087 | 13.7 | 1027 | 46.4 | 707 | 61.0 | 779 | 83.0 | 463 | 92.2 | 198 | 53.9 | 710 |
| CANON-Eff ($\alpha = 0.88$) | 27.3 | 816 | 15.3 | 862 | 43.9 | 582 | 59.3 | 649 | 84.4 | 386 | 91.4 | 166 | 53.6 | 577 |
| CANON-Eff ($\alpha = 0.96$) | 29.7 | 1216 | 19.0 | 1136 | 48.4 | 881 | 62.3 | 936 | 85.8 | 533 | 92.0 | 233 | 56.2 | 822 |

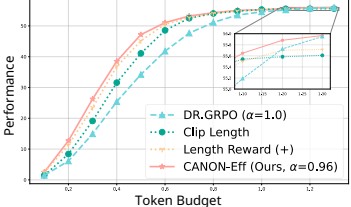

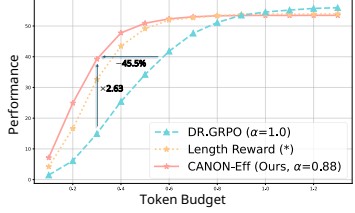

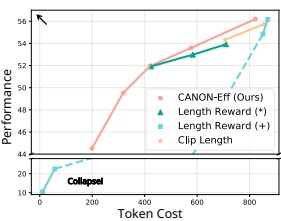

(a) CANON-Eff with $\alpha = 0.96$ consistently outperforms baselines methods.

(b) CANON-Eff with $\alpha = 0.88$ achieves significantly better performance at low token budgets.

(c) The Pareto frontier in the trade-off between performance and token efficiency.

Figure 4: Budget-Performance and Cost-Performance Curves for Efficient Reasoning. This figure compares the reasoning efficiency of CANON-Eff against baselines under various token budgets.

**Baselines.** We select three types of baseline methods towards efficient reasoning: (1) Clip Length that directly clips the maximum output length (Hou et al., 2025), (2) Length Reward (+) that adds length penalties terms in the training reward (Luo et al. (2025), $+\text{coeff} * (\frac{\text{mean}_{G_q}(L)}{L} - 1)$), and (3) Length Reward (*) that multiplies a normalized length coefficient on the reward (Arora & Zanette (2025), $*(1 - \text{coeff} * \text{sigmoid}(\frac{L - \text{mean}_{G_q}(L)}{\text{std}_{G_q}(L)})))$. All these baselines are conducted with DR.GRPO.

**CANON achieves better performance with shorter responses compared with baselines.** We present the detailed performance of the top-performing models for each method across various benchmarks in Table 3. CANON-Eff with $\alpha = 0.96$ Pareto dominates the results of Clip Length and Length Reward (+), reducing the length by 26.3% compared to DR.GRPO while only decreasing performance by 0.4 points. Figure 4 shows that CANON-Eff with $\alpha = 0.96$ consistently outperforms the baseline methods in both low-token-budget and high-token-budget scenarios. Since models trained with the Length Reward (*) exhibit significantly lower length with low performance at the same time, it is difficult to fairly compare with other baselines. To this end, we include an additional model trained with CANON-Eff with $\alpha = 0.88$ that has comparable performance. 4b indicates that CANON with $\alpha = 0.88$ shows better token efficiency compared with Length Reward (*), achieving 2.63 times the performance of DR.GRPO in low-token-budget scenarios, while reducing token consumption by 45.5% at the same performance level.

**CANON achieves a better Pareto frontier and stably explores the entire frontier.** To draw the cost-performance curves for each method, we draw the Pareto frontier of CANON-Eff with the results of $\alpha = 0.5, 0.7, 0.8, 0.88, 0.96$. For Length Clipping, we respectively present the results with maximum lengths of 2048 and 1024 in the Pareto frontier. For Length Reward (+), penalty coefficients of 0.001, 0.004, 0.005, and 0.1 are used, respectively. For Length Reward (*), we utilize the coefficients of 0.05, 0.2, and 0.4. 4c shows that all the frontier from baselines are dominated by the frontier of CANON-Eff's. It is noteworthy that after the coefficient of Length Reward (+) is adjusted from 0.004 to 0.005, its performance drops from 54.8 to 22.5. In contrast, CANON-Eff remains consistently stable, exploring the Pareto frontier efficiently.

# 6 ANALYSIS

In this section, we analyze how `CANON-Dynamic` and `CANON-Eff` effectively improve the task performance and reasoning efficiency.

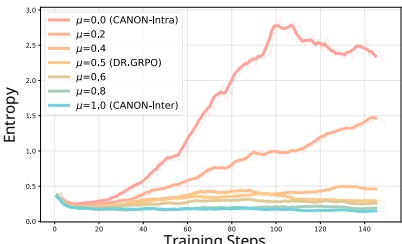

**Different advantage combinations of `CANON` select different trends of the target metrics.** Due to the different baseline rewards being compared, `CANON-Inter` tends to favor correct answers from the group with a higher average reward, while `CANON-Intra` selects correct answers from the group with a lower average reward. We compare the effects of `CANON` on their target metrics across seven

Figure 5: `CANON` shows hierarchical trends of target metrics through different combinations of `CANON-Inter` and `CANON-Intra`.

different settings, with $\mu$ ranging from 0.0 to 1.0. When entropy is considered, figure 5 shows that a larger $\mu$ (favoring more `CANON-Inter`) leads to a reduction in entropy, whereas a smaller $\mu$ (favoring more `CANON-Intra`) promotes an increase in entropy. The results demonstrates a hierarchical trend in the metric changes, indicating the effectiveness of controlling and selecting different trends from `CANON-Inter` and `CANON-Intra`. In this way, `CANON-Dynamic` can boost the task performance by adjusting different combinations of the two components.

**`CANON` can achieve positive gains of more rethinking and high training efficiency through scheduling of two advantages.** As shown in Figure 6, we record the performance genuinely brought by reflections and the curve of training reward. Although `CANON-Intra` achieves positive gains from more reflections, its training reward experiences a significant decline. In contrast, `CANON-Inter`, which shows a similar trend of DR.GRPO, has not yet achieved positive returns even by step 360, but maintains a higher training reward. `CANON-Dynamic`, on the other hand, not only achieves positive gains of rethinking but also makes a training reward on a par with `CANON-Inter`'s. This explains why `CANON-Dynamic` can achieve comprehensive leading performance in both math and complex logic reasoning tasks.

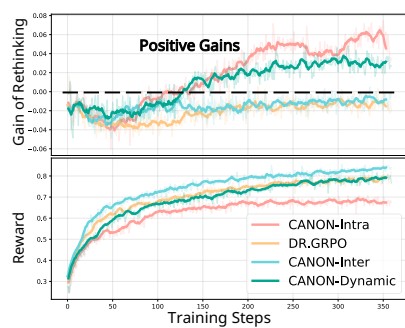

Figure 6: `CANON-Dynamic` with scheduled $\mu$ has positive gains of rethinking and high training score at the same time.

**`CANON` amplifies only the advantage attributable to the metric used for grouping, without amplifying the influence of other factors.** As shown in Table 4, directly scaling the advantage ($A = A * 2$) fails to improve performance the way CANON does. Any minor gains likely stem from faster learning progress due to an effectively larger learning rate, but this comes at the cost of degraded performance—particularly on out-of-domain logical reasoning tasks. This suggests that the key to `CANON`' success is not simply amplifying the advantage signal, but rather selectively amplifying specific signals, and that's why we introduce a regrouping operation.

Table 4: The performance comparison between the direct numerical amplification of advantage and `CANON`.

| Methods | Math | Logic |
|---|---|---|
| DR.GRPO | 55.7 | 26.2 |
| *Direct Numerical Amplification* | | |
| Numerical Scaling | 56.1 | 25.1 |
| Entropy Adv | 56.3 | 18.5 |
| *CANON* | | |
| `CANON-Intra` | 54.7 | **29.1** |
| `CANON-Inter` | **57.6** | 25.7 |

# 7 CONCLUSION

In this paper, we introduce `CANON`, a novel reinforcement learning framework for large reasoning models that leverages human priors on training metrics (e.g., entropy, response length) without presuming their directional impact on performance. Extensive experiments across six math reasoning benchmarks and three high-complexity logic reasoning tasks demonstrate that CANON significantly outperforms prior advantage estimation methods like DR.GRPO. `CANON` also supports flexible weighting of different metric trends, where `CANON` based on response length achieves a superior Pareto frontier in the performance-efficiency trade-off. Our analysis further confirms that `CANON` promotes beneficial behaviors such as effective exploration and reflection, which are critical for solving complex reasoning problems.

## ACKNOWLEDGMENTS

This work is supported by Shanghai Artificial Intelligence Laboratory.

## ETHICS STATEMENT

This work aims to introduce human priors about key metrics into reinforcement learning by proposing a novel advantage estimation framework named CANON, which amplifies the impact of target metrics without presuming preferences. The experiments in this paper are limited to reasoning tasks conducted on open-source models, datasets, and benchmarks, which will not raise ethical concerns. We hope to explore the potential of CANON to enhance the security of large language models in the future, thereby promoting their reliable and trustworthy development.

## REPRODUCIBILITY STATEMENT

We aim to include both the high-level and low-level details of our method in the setup paragraphs of Section 5 and Appendix C to reproduce our results. All experiments are conducted on open-source LLMs and benchmarks. We employ open-source datasets for the Qwen series LLMs, provide a detailed description of the prompts used for training and evaluation, and comprehensively present the construction process of the training dataset for the Llama series LLM. Our code implementation is based on VeRL (Sheng et al., 2024), which is applied with focused modifications in the advantage computation part, enhancing the reproducibility of our work. Please access our code base via the following anonymous link: CANON.

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

## A    LIMITATIONS.

Based on feasibility and motivation, this work focuses on conditions that can be specified through numerical ordering, without exploring conditions that are more complex and harder to verify. Due to limitations in paper length and computation resources, this work primarily conducts the CANON based on two metrics—response length and entropy—while other training metrics remain unexplored. Additionally, the paper considers only one metric at a time, without attempting to incorporate multiple metrics simultaneously. This demonstrates that the perspective and framework proposed in this work is flexible and hold significant potential for extension, which can be further explored in future research.

## B    THE USE OF LARGE LANGUAGE MODELS.

LLMs primarily assist this work in two aspects: on one hand, they are used for aiding our writing, and on the other hand, they sometimes serve as a coding assistant during the programming of our code base.

## C    EXPERIMENTS DETAILS.

### C.1    RETHINKING PATTERNS.

Following Gandhi et al. (2025), we firstly samples 10000 responses of Qwen3-32B Yang et al. (2025) and utilize the modified prompts from (Gandhi et al., 2025) to collect the rethinking patterns of verification, sub-goal setting, and backtracking. Then we match these patterns in a few Question-Answer instances and filter out overly frequent conjunctions, overly short words, and semantically ambiguous phrases. The number of remaining keywords and regular expressions is 334 for verification, 1036 for sub-goal setting, and 532 for backtracking.

### C.2    THE MAXIMUM TOKEN BUDGET SETUPS.

We set the maximum token budget for each benchmark based on its difficulty and the average token length observed from models trained with DR.GRPO, as shown in Figure 5. When plotting the performance-budget curve, we normalize the maximum token budget of each benchmark to 1.0. We then evaluate the performance of all benchmarks under token budgets ranging from 0.1× to 1.3× their respective maximum budget, averaging the results across benchmarks at each budget ratio and displaying them in the figure.

Table 5: Benchmark-wise Maximum Token Budget.

| Benchmark | Avg. Tokens (unlimited) | Max Token Budget |
|---|---|---|
| GSM8k | 349 | 600 |
| MATH-500 | 728 | 1500 |
| AMC | 1214 | 1800 |
| OlympiadBench | 1172 | 1800 |
| AIME 2024 | 1640 | 2000 |
| AIME 2025 | 1586 | 2000 |

### C.3    REASONS FOR EXPANDING THE CONTEXT WINDOW OF MODELS FROM QWEN2.5-MATH SERIES.

Initially, we uses the setting of Section 5.1; however, during the training process, too much length clipping (> 30%) results in nearly incomparable experimental outcomes, as shown in Figure 7. Therefore, we expand Qwen2.5-Math-7B's context limit from 4096 to 16384 and set the maximum output length to 8192, which alleviates this phenomenon.

### C.4    SYSTEM PROMPT.

For the training and inference of Qwen series models, we share the same system prompt as follows.

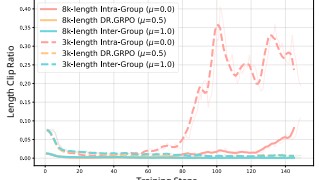
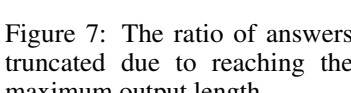
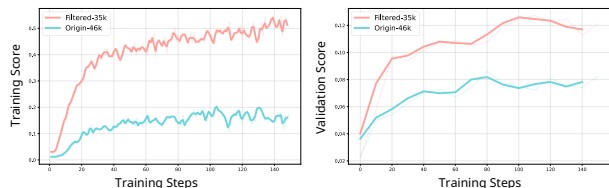

Figure 7: The ratio of answers truncated due to reaching the maximum output length.

Figure 8: The score curves of the training set and validation set from the newly constructed dataset with 35k data and the original dataset used for the Qwen series models, respectively.

> Your task is to follow a systematic, thorough reasoning process before providing the final solution. This involves analyzing, summarizing, exploring, reassessing, and refining your thought process through multiple iterations. Structure your response into two sections: Thought and Solution. In the Thought section, present your reasoning using the format: "`<think>`\n thoughts `</think>`\n". Each thought should include detailed analysis, brainstorming, verification, and refinement of ideas. After "`</think>`\n" in the Solution section, provide the final, logical, and accurate answer, clearly derived from the exploration in the Thought section. If applicable, include the answer in \boxed{} for closed-form results like multiple choices or mathematical solutions.

## C.5 Construction of training dataset for Llama3.1-8B.

Since the pretraining of Llama3.1-8B lacks data for long chain-of-thought and mathematical reasoning, its average training reward based on the original dataset used for Qwen2.5-Math remains below 0.2. To enhance training efficiency, we employ three Llama series models (Llama3.1-8B, Llama3.1-8B-Instruct, and Llama3.1-70B) to generate solutions for each problem across four datasets (training set of GSM8k (Cobbe et al., 2021), training set of MATH (Hendrycks et al., 2021), a 46k subset of OpenR1-Math-220k (Hu et al., 2025; Yan et al., 2025), and DeepMath-103k (He et al., 2025)). We then filter out questions whose accuracy of Pass@8 > 0, ultimately selecting 35k samples for training the Llama3.1-8B model. Concurrently, due to Llama3.1-8B's limited instruction-following capability, we simplify the output format requirements in its system prompt.

> Your task is to follow a systematic, thorough reasoning process before providing the final solution. This involves analyzing, summarizing, exploring, reassessing, and refining your thought process through multiple iterations. Structure your response into two sections: Thought and Solution. In the Thought section, each thought should include detailed analysis, brainstorming, verification, and refinement of ideas. In the Solution section, provide the final, logical, and accurate answer, clearly derived from the exploration in the Thought section. If applicable, include the answer in \boxed{} for closed-form results like multiple choices or mathematical solutions. Let's think step by step.

The training curves for this 35k dataset and the original 46k training dataset over 150 training steps are shown in the Figure 8. It demonstrates that Llama3.1-8B has significantly higher learning effectiveness on the newly constructed dataset.

## C.6 Scheduling strategies of coefficient to balance `CANON-INTER` and `CANON-INTRA`.

We try four different scheduling strategies and show the best of them for each model. Figure 9 shows the dynamics of $\mu$ in the training process from the *First-Inter-Later-Intra* ($\mu = 1 - \Lambda$) and *First-Intra-Later-Inter* ($\mu = \Lambda$). *Cosin-First-Inter-Later-Intra* and *Cosin-First-Intra-Later-Inter*

Table 6: Detailed experimental results of CANON on Llama3.1-8B and Qwen2.5-Math-1.5B . All models are evaluated under a unified setting. **Bold** and underline indicate the best and second-best results, respectively.

| Model | Math Reasoning | | | | | | | | High Complexity Reasoning | | | | |
|---|---|---|---|---|---|---|---|---|---|---|---|---|---|
| | AIME 24 | AIME 25 | Olympiad | AMC | MATH-500 | GSM8k | Tokens | Acc | Mid | Large | XLarge | Tokens | Acc |
| Qwen2.5-Math-1.5B | | | | | | | | | | | | | |
| DR.GRPO ($\mu = 0.5$) | 13.3 | **11.0** | **43.9** | 48.8 | 77.0 | 84.3 | 2381 | 46.4 | 23.7 | 9.7 | 5.0 | 9215 | 12.8 |
| CANON-Intra | 15.0 | 9.7 | 39.1 | 47.1 | 75.2 | **84.5** | 4092 | 45.1 | **27.7** | 11.5 | 4.9 | 11718 | 14.7 |
| CANON-Inter | 14.3 | 9.3 | 41.8 | 49.4 | **78.8** | 82.7 | **1876** | 46.1 | 23.1 | 9.6 | 5.8 | 8342 | 12.8 |
| CANON-Dynamic | **16.0** | 10.0 | 42.4 | **50.2** | 78.6 | 83.3 | 2479 | **46.8** | 27.0 | **16.3** | **7.9** | **7070** | **17.0** |
| Llama3.1-8B | | | | | | | | | | | | | |
| DR.GRPO ($\mu = 0.5$) | 1.3 | **0.3** | **8.3** | 11.3 | 32.0 | 78.9 | 9476 | 22.0 | 21.1 | 13.8 | 9.7 | 6370 | 14.9 |
| CANON-Intra | 1.0 | 0.0 | 7.7 | 10.2 | 27.2 | 78.9 | 23961 | 20.8 | 24.3 | 16.7 | 10.3 | 17753 | 17.1 |
| CANON-Inter | **2.7** | 0.0 | 8.0 | 10.0 | 31.6 | 79.8 | 3671 | 22.1 | 17.9 | 13.8 | 9.5 | **1331** | 13.7 |
| CANON-Dynamic | 0.7 | 0.0 | 7.1 | **12.4** | **33.8** | **81.4** | 2354 | **22.6** | **26.0** | **18.4** | **12.3** | 1685 | **18.9** |

schedule the value of $\mu$ with a cosine annealing function $\Psi$ with restarts and warm-up:

$$
\Psi = \begin{cases} \mu_{\max} \cdot \dfrac{s+1}{w} & \text{if } s < w \\ \mu_{\min} + \dfrac{1}{2}(\mu_{\max} - \mu_{\min})\left(1 + \cos\left(\pi \cdot \dfrac{s'}{\left\lfloor \frac{S-w}{c} \right\rfloor}\right)\right) & \text{if } s \geq w \text{ and } s' = s - w \bmod \left\lfloor \frac{S-w}{c} \right\rfloor \end{cases},
$$

$$(10)$$

where $c$ denotes the number of restart and $w$ is the warm-up step. $s$ is the current step of training and $S$ is the total step. $\mu_{\max}$ and $\mu_{\min}$ denote the specified maximum and minimum values of $\mu$. We use $c = 3$, $w = 30$ and $S = 150$ for both strategies.

In strategy *Cosin-First-Inter-Later-Intra*, we utilize $\mu = \Psi$ with $\mu_{\max} = 1.0$ and $\mu_{\min} = 0.4$, respectively, while in strategy *Cosin-First-Intra-Later-Inter*, we utilize $\mu = 1 - \Psi$) with $\mu_{\max} = 0.6$ and $\mu_{\min} = 0.0$, respectively. The changes in $\mu$ under these strategies are shown in the Figure 10. Ultimately, based on training performance, we selected strategy *Cosin-First-Inter-Later-Intra* for Qwen2.5-7B and Llama, and strategy *First-Inter-Later-Intra* for Qwen2.5-1.5B.

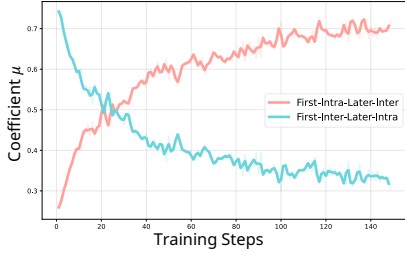

Figure 9: The changes of $\mu$ for two scheduling strategies based on accuracy during training.

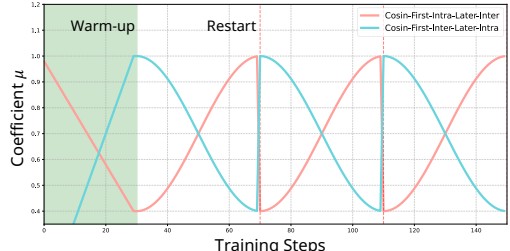

Figure 10: The changes of $\mu$ for two scheduling strategies based on training steps during training.

## C.7 DETAILED EXPERIMENTAL RESULTS ON LLAMA3.1-8B AND QWEN2.5-MATH-1.5B.

Here we show the detailed test results for Llama3.1-8B and Qwen2.5-Math-1.5B for the comparison between CANON, CANON-Dynamic and its baseline. As shown in Table 6, consistent with Qwen2.5-Math-7B, Llama3.1-8B achieves superior performance on math tasks with CANON-Inter and leads on logic tasks with CANON-Intra, while CANON-Dynamic outperforms the baseline across both tasks. On Qwen2.5-Math-1.5B, CANON-Inter does not achieve a lead in math performance; however, its dynamic variant CANON-Dynamic still surpasses the baseline in both tasks, demonstrating the effectiveness of CANON.

# D    ADDITIONAL EXPERIMENTS.

## D.1    CANON BASED ON ANOTHER METRIC.

To further verify that CANON remains effective under other grouping criteria, we conduct new experiments that use the number of per-token reflection steps in each response as the grouping metric. As shown in Table 7, it exhibits a trend similar to entropy-based CANON. Although both are slightly inferior to the entropy-based CANON, CANON-Inter still outperforms the baselines on mathematical reasoning, and CANON-Intra achieves better performance than the baselines on complex logic reasoning. Figure 11 demonstrate that CANON-Inter favors less-reflection responses and CANON-Intra encourages more reflections.

Table 7: Overall performance of CANON based on Per-token Reflection for Qwen2.5-Math-7B. All models are evaluated under a unified setting. **Bold** and underline indicate the best and second-best results, respectively.

| Model | Math Reasoning | | | | | | | | High Complexity Reasoning | | | | |
|---|---|---|---|---|---|---|---|---|---|---|---|---|---|
| | AIME 24 | AIME 25 | Olympiad | AMC | MATH-500 | GSM8k | Tokens | Acc | Mid | Large | XLarge | Tokens | Acc |
| DR.GRPO ($\mu = 0.5$) | 27.7 | **20.3** | 48.4 | 63.4 | 83.2 | 91.1 | 1522 | 55.7 | 39.2 | 24.4 | 15.1 | 4896 | 26.2 |
| Our Methods (Conditional Groups based on *Per-token Reflection*) | | | | | | | | | | | | | |
| CANON-Intra | 25.0 | 16.0 | 50.4 | 62.8 | 85.4 | 91.3 | 1912 | 55.1 | 41.0 | 25.5 | 15.5 | 4834 | 27.3 |
| CANON-Inter | 26.7 | 18.3 | 51.9 | 65.4 | 85.4 | 92.2 | 1739 | 56.6 | 37.4 | 17.0 | 8.5 | 7835 | 21.0 |

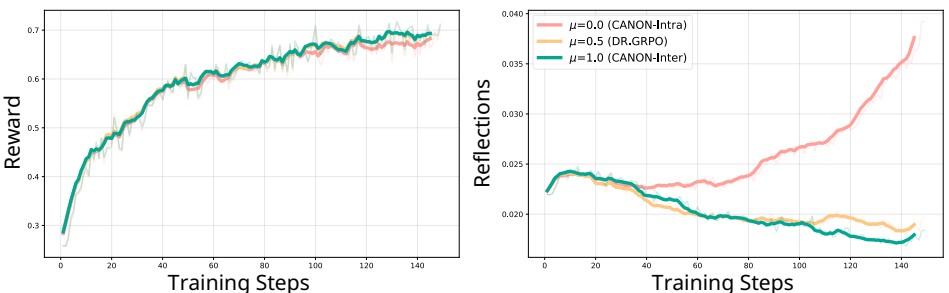

Figure 11: The training dynamic of CANON based on per-token reflection.

## D.2    OTHER ALTERNATIVE SCHEDULING STRATEGIES OF CANON-DYNAMIC

We conduct further experiments that utilize other alternative scheduling strategies that were used to be performed in the scheduler of learning rate, including the *Lambda strategy* (PyTorch, 2025) and *Cyclic-triangular2 strategy* (Smith, 2017). Following the setting of tried scheduling strategies, both of these strategies schedule $\mu$ from 1.0 down to 0.4.

The new experimental results in Table 8 show that Lambda strategy achieves slightly better performance than DR.GRPO on math tasks but performs worse on logic reasoning tasks. This may be because they fail to sufficiently leverage intra-group advantages in the later stages of training. This experiment demonstrates the rationale behind CANON's tried scheduling strategies and highlights the practical flexibility of the CANON framework.

## D.3    DYNAMIC SCHEDULING ON LENGTH-BASED CANON.

When we consider response length, CANON-Inter tends to produce shorter responses, whereas CANON-Intra favors longer ones. However, these trends in response length do not translate into performance gains. This is precisely why we only applied dynamic scheduling to the entropy-based variant of CANON. To prove this, we try dynamic scheduling on length-based CANON under the setting of entropy-based CANON—initially favoring shorter responses and gradually shifting toward longer ones as training progresses. The results are shown below. Although the responses are shorter than those of DR.GRPO, the math performance slightly declines, as shown in Table 9. While this

Table 8: Overall performance of `CANON-Dynamic` based on other two scheduling strategies for Qwen2.5-Math-7B. All models are evaluated under a unified setting. **Bold** and underline indicate the best and second-best results, respectively.

| Model | Math Reasoning | | | | | | | | High Complexity Reasoning | | | | |
|---|---|---|---|---|---|---|---|---|---|---|---|---|---|
| | AIME 24 | AIME 25 | Olympiad | AMC | MATH-500 | GSM8k | Tokens | Acc | Mid | Large | XLarge | Tokens | Acc |
| DR.GRPO ($\mu = 0.5$) | 27.7 | 20.3 | 48.4 | 63.4 | 83.2 | 91.1 | 1522 | 55.7 | 39.2 | 24.4 | 15.1 | 4896 | 26.2 |
| | | | | | | CANON-Dynamic based on *Entropy* | | | | | | | |
| *Cosin-First-Inter-Later-Intra* | **30.0** | 17.7 | 50.7 | 63.3 | **86.6** | 91.8 | 1452 | 56.7 | 40.4 | **30.5** | 16.6 | 3535 | **29.2** |
| *First-Inter-Later-Intra* | 28.0 | 20.3 | **52.4** | **64.6** | 84.2 | **92.6** | **1328** | **57.0** | **41.7** | 26.6 | 16.5 | 3862 | 28.3 |
| *Cyclic-triangular2 strategy* | 24.0 | 18.0 | 49.3 | 63.3 | 84.8 | 91.3 | 1647 | 55.1 | 37.4 | 22.1 | 14.5 | 5203 | 24.6 |
| *Lambda strategy* | 26.0 | **22.0** | 49.3 | 63.5 | 85.2 | 91.5 | 1744 | 56.3 | 37.1 | 21.8 | 13.6 | 5297 | 24.1 |

method shows improved performance on complex reasoning tasks, it still does not surpass either CANON-Inter or CANON-Intra based on length.

Table 9: Overall performance of `CANON-Dynamic` based on length response for Qwen2.5-Math-7B. All models are evaluated under a unified setting. **Bold** and underline indicate the best and second-best results, respectively.

| Model | Math Reasoning | | | | | | | | High Complexity Reasoning | | | | |
|---|---|---|---|---|---|---|---|---|---|---|---|---|---|
| | AIME 24 | AIME 25 | Olympiad | AMC | MATH-500 | GSM8k | Tokens | Acc | Mid | Large | XLarge | Tokens | Acc |
| DR.GRPO ($\mu = 0.5$) | 27.7 | 20.3 | 48.4 | 63.4 | 83.2 | 91.1 | 1522 | 55.7 | 39.2 | 24.4 | 15.1 | 4896 | 26.2 |
| | | | | | | CANON based on *Length* | | | | | | | |
| CANON-Intra | 21.7 | 19.0 | 49.9 | 63.0 | 86.2 | **92.2** | 2176 | 55.3 | 41.8 | 25.6 | 14.7 | 4364 | 27.4 |
| CANON-Inter | 27.3 | 19.3 | 47.6 | 64.2 | 82.6 | 91.0 | **1008** | 55.3 | **42.7** | 28.6 | 17.1 | 3652 | **29.5** |
| CANON-Dynamic | 27.7 | 17.7 | 48.3 | 63.6 | 84.6 | 91.7 | 1393 | 55.6 | 39.6 | 24.7 | 17.8 | 4333 | 27.3 |

## D.4 ANALYSIS OF $\mu$'S HYPERPARAMETER TUNING COMPLEXITY

Although `CANON-Dynamic` introduces a hyperparameter $\mu$ to balance exploitation and exploration, unlike conventional regularization coefficients, $\mu$ carries rich physical meaning and does not add significant complexity. When $\mu$ equals 0.5, DR.GRPO achieves the simplest form of balance through a static weighted average. This observation inspired us: if a more dynamic balancing mechanism exists, it is natural that this method could outperform DR.GRPO—this is precisely why `CANON-Dynamic` works.

To analyze the hyperparameter tuning complexity re-introduced by $\mu$, we train Qwen2.5-Math-7B with a 4K context length using entropy-based `CANON`, showing how model performance and entropy vary with $\mu$. Table 10 indicates that, as $\mu$ increases from 0 (`CANON-Intra`) to 1 (`CANON-Inter`), in-domain mathematical performance steadily improves, out-of-domain logical reasoning performance gradually declines, and entropy consistently decreases—revealing a clear trend. Therefore, introducing $\mu$ does not increase the difficulty of hyperparameter tuning; rather, it extends the `CANON` framework and offers new insights on how we can utilize `CANON`.

Table 10: Performance and entropy across different $\mu$ values.

| $\mu$ | 0.0 | 0.2 | 0.4 | 0.5 | 0.6 | 0.8 | 1.0 |
|---|---|---|---|---|---|---|---|
| Math | 54.2 | 54.9 | 56.0 | 56.6 | 56.7 | 56.4 | **57.9** |
| Logic | **27.1** | 25.1 | 24.6 | 23.8 | 25.6 | 22.9 | 22.5 |
| Entropy | 2.40 | 1.28 | 0.46 | 0.39 | 0.26 | 0.19 | 0.15 |

## D.5 ANALYSIS OF $\alpha$'S HYPERPARAMETER TUNING COMPLEXITY

Insights from `CANON` reveal that the choice of metric trend primarily occurs in the inter-group advantage computation. Therefore, in scenarios where inference efficiency is desired, we only need to slightly reduce the reward weight for the long-response group in `CANON-Inter`, prompting the model to favor shorter answers. Meanwhile, since `CANON-Intra` remains unchanged, `CANON-Eff` fully preserves the model's exploration capability, achieving a superior performance–efficiency Pareto frontier. The hyperparameter $\alpha$ introduced in CANON-Eff not only allows flexible tuning

toward specific application needs and thorough exploration of the Pareto frontier, but also ensures greater stability and smoother behavior compared to baselines (like the collapse of Length Reward (+)) —because it minimally alters the training process (only modifying the inter-group advantage computation).

To analyze the hyperparameter tuning complexity re-introduced by $\alpha$. We show the detailed performance and token length of these CANON-Eff models. Table **??** indicates that as $\alpha$ gradually decreases from 0.96 to 0.5, model performance declines modestly, while the number of tokens consumed drops significantly—again aligning with our understanding. Therefore, in efficient reasoning tasks, introducing $\alpha$ not only avoids increasing hyperparameter tuning difficulty but also enables users to apply CANON-Eff more flexibly according to their specific needs.

Table 11: Performance and token cost across different $\alpha$ values.

| $\alpha$ | 0.5 | 0.7 | 0.8 | 0.88 | 0.96 |
|---|---|---|---|---|---|
| Performance | 44.5 | 49.5 | 52.0 | 53.6 | **56.2** |
| Token Cost | **198.9** | 317.9 | 420.6 | 576.6 | 822.4 |

### D.6  CANON SELECTS APPROPRIATE METRICS AS THE TARGET.

We conduct a simple ablation study on the target metrics considered by CANON. As shown in Table 12, random regrouping achieves only the same performance as the baseline method while producing longer responses, thus failing to improve either performance or efficiency compared to the baseline. In contrast, CANON-Inter based on the response length excels in the token efficiency with 33.8% shorter responses, and the entropy-based CANON-Inter delivers the best performance (57.6 points) among the comparisons.

Table 12: The accuracy and token cost of CANON-Inter with different metrics.

| Methods | Acc | Tokens |
|---|---|---|
| DR.GRPO | 55.7 | 1522 |
| Random regrouping | 55.7 | 1557 |
| CANON-Inter | | |
| based on *Length* | 55.3 | **1008** |
| based on *Entropy* | **57.6** | 1466 |

## E  DETAILED DERIVATION OF THEOREM 1 AND 2

**Theorem 1** (Situations with clearer advantage signal). *Suppose that condition c is based on numerical comparisons and can be derived through sorting of metrics. Further assume that the sampled response o to query q satisfy condition c with probability $p \in (0, 1)$, and $\mathbf{E}_{o \ satisfy \ c}[R_o] \neq \mathbf{E}_{o \ not \ satisfy \ c}[R_o]$. Then, we have:*

$$\frac{|\hat{A}^{inter}_{q,o,t}|}{|\hat{A}^{DR.GRPO}_{q,o,t}|} > 1, \ only \ when \ |C^+_q| = |C^-_q| \ if \ |C^+_q| \ is \ a \ constant. \tag{11}$$

*Proof of Theorem 1.* Given a prompt $q$, the set of all responses that satisfy condition $c$ can be denoted as $\mathcal{C}$. We use $p = \mathrm{P}(o \in \mathcal{C}|q, \theta) \in (0, 1)$ to describe the probability that a response $o$ satisfying condition $c$ is provided to the prompt $q$ by an LLM with parameter $\theta$. Assuming that when condition $c$ is satisfied, the probability of the correct response is $a_+$, and when condition $c$ is not satisfied, the probability of the correct response is $a_-$. Denoting the correctness of the response $o$ to query $q$ as $R_o$, then we have:

$$\mathbf{E}_{o \in \mathcal{C}}[R_o] = a_+ \ and \ \mathbf{E}_{o \notin \mathcal{C}}[R_o] = a_- \ . \tag{12}$$

### E.1  DR.GRPO

Sampling a group of responses $G_q$ to the prompt $q$, the advantage $\hat{A}^{DR.GRPO}_{q,o,t}$ of a response $o$ can be calculated as:

$$\hat{A}^{DR.GRPO}_{q,o,t} = R_o - \mathrm{mean}(\{R_{o'}|o' \in G_q\}). \tag{13}$$

We use $A^{\text{DR.GRPO}}(o, c)$ to denote the **average** advantage of the responses that **satisfy** condition $c$, and utilize $\tilde{A}^{\text{DR.GRPO}}(o, c)$ to describe the average advantage of the other responses that **do not satisfy** condition $c$.

$$
\begin{aligned}
A^{\text{DR.GRPO}}(o, c) &= \mathbf{E}_{o \in \mathcal{C}}[\hat{A}^{\text{DR.GRPO}}_{q,o,t}] \\
&= \mathbf{E}_{o \in \mathcal{C}}[R_o] - \mathbf{E}_{o \in G_q}[R_o] \\
&= a_+ - [\mathrm{P}(o \in \mathcal{C}|q, \theta)\mathbf{E}_{o \in \mathcal{C}}[R_o] + \mathrm{P}(o \notin \mathcal{C}|q, \theta)\mathbf{E}_{o \notin \mathcal{C}}[R_o]] \\
&= a_+ - pa_+ - (1-p)a_- = (a_+ - a_-)(1-p) ,
\end{aligned}
\tag{14}
$$

$$
\tilde{A}^{\text{DR.GRPO}}(o, c) = (a_- - a_+)p .
\tag{15}
$$

### E.2 INTER-GROUP ADVANTAGE (CANON-INTER)

We sort the sampled responses based on the numerical value considered by condition $c$, and split them at position $k$ into two groups. Based on the symmetry of the inter-group advantage, we can denote these $k$ responses as $C_q^+$. We use $\lambda := \frac{|C_q^+|}{|G_q|}$ to simplify the notation, and denote the average inter-group advantage with $A_\lambda(o, c, p)$ for the responses that **satisfy** condition $c$. $\tilde{A}_\lambda(o, c, p)$ is utilized to represent the average inter-group advantage of those responses that **do not satisfy** condition $c$.

Then, we can compute the average reward of each group as follows.

$$
\begin{aligned}
\mathbf{E}_{o \in C_q^+}[R_o] &= [\mathrm{P}(o \in \mathcal{C}|q, \theta, o \in C_q^+)\mathbf{E}_{o \in \mathcal{C}}[R_o] + \mathrm{P}(o \notin \mathcal{C}|q, \theta, o \in C_q^+)\mathbf{E}_{o \notin \mathcal{C}}[R_o]] \\
&= \begin{cases} \frac{p}{\lambda}\mathbf{E}_{o \in \mathcal{C}}[R_o] + \frac{\lambda - p}{\lambda}\mathbf{E}_{o \notin \mathcal{C}}[R_o], \text{if } \lambda \geq p \\ \\ \mathbf{E}_{o \in \mathcal{C}}[R_o], \text{if } \lambda < p \end{cases} \\
&= \begin{cases} \frac{p}{\lambda}a_+ + \frac{\lambda - p}{\lambda}a_-, \text{if } \lambda \geq p \\ \\ a_+, \text{if } \lambda < p \end{cases} ,
\end{aligned}
\tag{16}
$$

$$
\begin{aligned}
\mathbf{E}_{o \notin C_q^+}[R_o] &= [\mathrm{P}(o \in \mathcal{C}|q, \theta, o \notin C_q^+)\mathbf{E}_{o \in \mathcal{C}}[R_o] + \mathrm{P}(o \notin \mathcal{C}|q, \theta, o \notin C_q^+)\mathbf{E}_{o \notin \mathcal{C}}R_o] \\
&= \begin{cases} \mathbf{E}_{o \notin \mathcal{C}}[R_o], \text{if } \lambda \geq p \\ \\ \frac{p-\lambda}{1-\lambda}\mathbf{E}_{o \in \mathcal{C}}[R_o] + \frac{1-p}{1-\lambda}\mathbf{E}_{o \notin \mathcal{C}}[R_o], \text{if } \lambda < p \end{cases} \\
&= \begin{cases} a_-, \text{if } \lambda \geq p \\ \\ \frac{p-\lambda}{1-\lambda}a_+ + \frac{1-p}{1-\lambda}a_-, \text{if } \lambda < p \end{cases} .
\end{aligned}
\tag{17}
$$

Therefore, we can calculate the average advantages:

$$
\begin{aligned}
A_\lambda(o, c, p) &= \mathbf{E}_{o \in \mathcal{C}}[R_o - \mathrm{P}(o \in C_q^+|q, \theta, o \in \mathcal{C})\mathbf{E}_{o' \notin C_q^+}[R_{o'}] - \mathrm{P}(o \notin C_q^+|q, \theta, o \in \mathcal{C})\mathbf{E}_{o' \in C_q^+}[R_{o'}]] \\
&= \mathbf{E}_{o \in \mathcal{C}}[R_o] - \begin{cases} a_-, \text{if } \lambda \geq p \\ \frac{\lambda}{p}[\frac{p-\lambda}{1-\lambda}a_+ + \frac{1-p}{1-\lambda}a_-], \text{if } \lambda < p \end{cases} - \begin{cases} 0, \text{if } \lambda \geq p \\ \frac{p-\lambda}{p}a_+, \text{if } \lambda < p \end{cases} \\
&= \begin{cases} a_+ - a_-, \text{if } \lambda \geq p \\ \frac{\lambda(1-p)}{p(1-\lambda)}(a_+ - a_-), \text{if } \lambda < p \end{cases} ,
\end{aligned}
\tag{18}
$$

$$
\begin{aligned}
\tilde{A}_\lambda(o, c, p) &= \mathbf{E}_{o \notin \mathcal{C}}[R_o - \mathrm{P}(o \in C_q^+|q, \theta, o \notin \mathcal{C})\mathbf{E}_{o' \notin C_q^+}[R_{o'}] - \mathrm{P}(o \notin C_q^+|q, \theta, o \notin \mathcal{C})\mathbf{E}_{o' \in C_q^+}[R_{o'}]] \\
&= \mathbf{E}_{o \notin \mathcal{C}}[R_o] - \begin{cases} \frac{\lambda - p}{1-p}a_-, \text{if } \lambda \geq p \\ 0, \text{if } \lambda < p \end{cases} - \begin{cases} \frac{1-\lambda}{1-p}[\frac{p}{\lambda}a_+ + \frac{\lambda-p}{\lambda}a_-], \text{if } \lambda \geq p \\ a_+, \text{if } \lambda < p \end{cases} \\
&= \begin{cases} \frac{p(1-\lambda)}{\lambda(1-p)}(a_- - a_+), \text{if } \lambda \geq p \\ a_- - a_+, \text{if } \lambda < p \end{cases} .
\end{aligned}
\tag{19}
$$

### E.3 COMPARISON

We have the ratio between inter-group advantage and DR.GRPO:

$$\frac{|A_\lambda(o,c,p)|}{|A^{\text{DR.GRPO}}(o,c)|} = \begin{cases} \frac{1}{1-p} > 1 & \text{if } \lambda \geq p \\ \frac{\lambda}{(1-\lambda)p} & \text{if } \lambda < p \end{cases}, \tag{20}$$

and

$$\frac{|\tilde{A}_\lambda(o,c,p)|}{|\tilde{A}^{\text{DR.GRPO}}(o,c)|} = \begin{cases} \frac{1-\lambda}{\lambda(1-p)} & \text{if } \lambda \geq p \\ \frac{1}{p} > 1 & \text{if } \lambda < p \end{cases}. \tag{21}$$

To accentuate the impact of a specific condition on advantages, the following is required:

$$\frac{1-\lambda}{\lambda(1-p)} > 1 \text{ if } \lambda \geq p, \text{ and } \frac{\lambda}{(1-\lambda)p} > 1 \text{ if } \lambda < p. \tag{22}$$

Then we have

$$\lambda < \frac{1}{2-p} \text{ if } \lambda \geq p, \text{ and } \lambda > \frac{p}{1+p} \text{ if } \lambda < p. \tag{23}$$

If $|C_q^+|$ is a constant, $\lambda$ is also a constant. Due to $\frac{1}{2-p} > \frac{1}{2}$ and $\frac{p}{1+p} < \frac{1}{2}$, $\lambda$ needs to satisfy $\lambda \leq \frac{1}{2}$ and $\lambda \geq \frac{1}{2}$ at the same time, consequently restricting the value of $\lambda$ to 0.5. In this way, we have $\frac{|C_q^+|}{|C_q^+|+|C_q^-|} = 0.5$, and finally $|C_q^+| = |C_q^-|$ $\qquad\square$

**Theorem 2** (Selective amplification based on specific metrics (proved in Appendix E)). *Consider independent conditions $c_1$ and $c_2$, and their corresponding sets $C_1$ and $C_2$ (i.e., $P(o \in C_1 \cap C_2|q,\theta) = P(o \in C_1|q,\theta)P(o \in C_2|q,\theta)$). When we fix the condition $c_1$, then for any value of the performance gap $a_{2+}$, $a_{2-}$ and the probability $P(o \in C_2|q,\theta)$ that induced by whether $c_2$ is satisfied, we have*

$$\frac{|\hat{A}_{q,o,t}^{\text{inter based on } c_1}|}{|\hat{A}_{q,o,t}^{\text{DR.GRPO}}|} \text{ is a constant.} \tag{24}$$

*which says* CANON *based on the condition $c_1$ will not amplify the influence of another independent condition $c_2$.*

*Proof of Theorem 2.* According to Eq. 20 and 21, the scaling factor depends only on the probability $p_1 = P(o \in C_1|q,\theta)$ that a response $o$ satisfying condition $c_1$ is provided to the prompt $q$ by an LLM with parameter $\theta$. Therefore, any irrelevant condition $c_2$ and its associated parameters cannot affect this ratio. $\qquad\square$

