# OpenReview forum: "Conditional Advantage Estimation for Reinforcement Learning in Large Reasoning Models"
_ICLR.cc/2026/Conference — ICLR 2026 Poster_

### Official Review · Reviewer_pJ7Q · 2025-10-29

**Soundness:** 3
**Presentation:** 3
**Contribution:** 3
**Rating:** 6
**Confidence:** 4

**Summary:**

This paper introduces CANON, an advantage estimation method for RLVR. Instead of relying on hand-crafted human priors such as the assumption that lower entropy is better or that higher entropy is better, CANON dynamically identifies which metric trend leads to better performance. It works by conditionally partitioning sampled responses into two equal-sized groups based on a target metric, then computing two types of advantages: CANON-Inter (comparing against the other group) and CANON-Intra (comparing within its own group). The overall advantage is computed as $\mu$ × CANON-Inter + (1 - $\mu$) × CANON-Intra. The authors find that CANON-Inter works better for math tasks while CANON-Intra works better for complex logic tasks. CANON-Dynamic schedules $\mu$ during training based on training accuracy or training steps, achieving a balance between the two approaches.

**Strengths:**

- CANON selects its own direction by empirically determining which metric trend performs better, rather than rolling with a fixed assumption. The motivation to remove hand-crafted directional priors and let the data determine which metric trend is beneficial appears novel and well-justified.
- The method provides a systematic way to control the emphasis on exploration (favored by CANON-Intra) and exploitation (favored by CANON-Inter). This offers a new approach to balance a metric, which is especially valuable for entropy that benefits from careful, dynamic balancing.

**Weaknesses:**

- While adaptive scheduling makes sense for entropy (balancing exploration and exploitation), its value is less clear for response length, where shorter responses are almost always preferable for efficiency.
- The choice between CANON-Dynamic and cosine-based scheduling appears to depend on the specific model being trained, with no universally optimal approach, thus requiring trial and error to determine the best strategy.
- This re-introduces hyperparameter tuning complexity, as the values of mu and alpha (group proportion parameter), along with their scheduling and adjustments, constitute additional hyperparameters that require careful calibration, which contradicts the paper's stated motivation.
- The method only considers one metric at a time, and it remains unclear what other metrics beyond entropy and length could be meaningfully employed with this approach.
- The evaluation metric for the AIME benchmarks, Avg@10, seems insufficient given their small sample size. To my knowledge, Avg@64 is more commonly used in practice to achieve stable and reliable results for such tasks.

**Questions:**

- Is there a principled way to incorporate multiple metrics simultaneously within a single unified advantage calculation?
- In the upper right corner of Figure 1, wouldn't it be clearer if the inter- and intra-group advantages were each visually subdivided to show their two constituent subgroups? This would help readers more readily understand that each advantage metric is calculated using two distinct subgroups.

---

> ### Author Response · Authors · 2025-11-20
> **Response to Reviewer pJ7Q  (Part Ⅰ)**
>
> Thank you for your great efforts in reviewing this paper. We will try our best to answer all your questions. **Please let us know if you still have further concerns, or if you are not satisfied with the current responses, so that we can further update the response ASAP.**
>
> ---
>
> **Q1:** "While adaptive scheduling makes sense for entropy (balancing exploration and exploitation), its value is less clear for response length, where shorter responses are almost always preferable for efficiency."
>
> **A1**: Thank you for your comment. In fact, this is tied to the objective of the specific task. When grouping by response length, our typical goal is to improve reasoning efficiency, and thus favoring shorter responses often genuinely serves that purpose. However, **“shorter is better” alone does not guarantee maintained performance while reducing length.** CANON offers a novel perspective on this issue: the choice of trend is applied specifically within the inter-group advantage component. Inspired by this, CANON-Eff reduces the weight for longer responses only in the inter-group advantage, while preserving the intra-group advantage computation—thereby shortening response length while retaining the model’s capacity for exploration.
>
> Meanwhile, when we consider response length, CANON-Inter tends to produce shorter responses, whereas CANON-Intra favors longer ones. However, these trends in response length do not translate into performance gains. This is precisely why we only applied dynamic scheduling to the entropy-based variant of CANON. To prove this, **we have followed your suggestions to conduct new experiments that** try dynamic scheduling on length-based CANON under the setting of entropy-based CONON—initially favoring shorter responses and gradually shifting toward longer ones as training progresses. The results are shown below. Although the responses are shorter than those of DR.GRPO, the math performance slightly declines. While this method shows improved performance on complex reasoning tasks, it still does not surpass either CANON-Inter or CANON-Intra based on length. We will include the new experimental results in the appendix.
>
> |                                 | Math     | Length   | Logic    | Length   |
> | ------------------------------- | -------- | -------- | -------- | -------- |
> | DR.GRPO                         | **55.7** | 1522     | 26.2     | 4896     |
> | CANON-Intra based on *Length*   | 55.3     | 2176     | 27.4     | 4364     |
> | CANON-Inter based on *Length*   | 55.3     | **1008** | **29.5** | **3652** |
> | CANON-Dynamic based on *Length* | 55.6     | 1393     | 27.3     | 4333     |
>
> ---
>
>
>
> **Q2:** "The choice between CANON-Dynamic and cosine-based scheduling appears to depend on the specific model being trained, with no universally optimal approach, thus requiring trial and error to determine the best strategy."
>
> **A2**: Thank you. As stated in Lines 334-342 of the original manuscript, we try four approaches and apply a fixed optimal schedule approach for each given model. Though our preliminary exploration reveals that different specific scheduling strategies can be beneficial for training models of varying sizes, we identify a single unified strategy that consistently outperforms DR-GPPO across all tested models and tasks, as shown below. We have **followed your suggestions to add the detailed data of different strategies as follows**. The experimental results indicate that ***First-Inter-Later-Intra*** **consistently performs better than DR.GRPO across three models and two tasks,** though its suboptimal performance on Qwen2.5-Math-7B and Llama 3.1-8B compared with *Cosin-First-Inter-Later-Intra.* Moreover, different models may have different numbers of parameters and different levels of capability. A specifically designed strategy is acceptable for better performance in practice. We will polish the presentation of this part in our revision and include these detailed experimental results in our revision.
>
> |          | Math     | Logic    |
> | ---- | -------- | -------- |
> | Qwen2.5-Math-7B                                |          |          |
> | DR.GRPO                                        | 55.7     | 26.2     |
> | CANON-Dynamic w\ Cosin-First-Inter-Later-Intra | 56.7     | **29.2** |
> | CANON-Dynamic w\ First-Inter-Later-Intra       | **57.0** | 28.3     |
> |                                                |          |          |
> | Qwen2.5-Math-1.5B                              |          |          |
> | DR.GRPO                                        | 46.4     | 12.8     |
> | CANON-Dynamic w\ Cosin-First-Inter-Later-Intra | **46.9** | 10.8     |
> | CANON-Dynamic w\ First-Inter-Later-Intra       | 46.8     | **17.0** |
> |      |          |          |
> | Llama 3.1-8B     |          |          |
> | DR.GRPO          | 22.0     | 14.9     |
> | CANON-Dynamic w\ Cosin-First-Inter-Later-Intra | **22.6** | **18.9** |
> | CANON-Dynamic w\ First-Inter-Later-Intra       | 22.1     | 17.7     |
>
> ---

---

> ### Author Response · Authors · 2025-11-20
> **Response to Reviewer pJ7Q  (Part Ⅱ)**
>
> **Q3:** "This re-introduces hyperparameter tuning complexity, as the values of mu and alpha (group proportion parameter), along with their scheduling and adjustments, constitute additional hyperparameters that require careful calibration, which contradicts the paper's stated motivation."
>
> **A3**: Thank you for your suggestions.
>
> > ... which contradicts the paper's stated motivation.
>
> The core CANON contribution—CANON‑Inter and CANON‑Intra—already achieves our stated goal: amplifying metric impact without presupposing direction, thereby eliminating the hand‑crafted penalty/reward coefficients that bias prior methods. These two variants alone outperform baselines on their respective strong suits (math for Inter, complex logic for Intra) using no additional hyperparameters beyond the regrouping itself.
>
> The two approaches with additional hyperparameters ($\mu$ for CANON-Dynamic and $\alpha$ for CANON-Eff) can be viewed as two application cases designed for improving performance and token efficiency of LLMs, respectively, inspired by the perspective given by CANON.
>
> > Analysis of $\mu$'s hyperparameter tuning complexity
>
> Although CANON-Dynamic introduces a hyperparameter $\mu$ to balance exploitation and exploration, unlike conventional regularization coefficients, $\mu$ carries rich physical meaning and does not add significant complexity. When $\mu$ equals 0.5, DR.GRPO achieves the simplest form of balance through a static weighted average. This observation inspired us: if a more dynamic balancing mechanism exists, it is natural that this method could outperform DR.GRPO—this is precisely why CANON-Dynamic works.
>
> We **have followed your suggestion to conduct new experiments that demonstrate the hyperparameter tuning complexity re-introduced by $\mu$.** We train Qwen2.5-Math-7B with a 4K context length using entropy-based CANON, showing how model performance and entropy vary with $\mu$. All other settings are identical to those in Lines 247–268 of the original manuscript. As $\mu$ increases from 0 (CANON-Intra) to 1 (CANON-Inter), in-domain mathematical performance steadily improves, out-of-domain logical reasoning performance gradually declines, and entropy consistently decreases—revealing a clear trend. Therefore, introducing $\mu$ does not increase the difficulty of hyperparameter tuning; rather, it extends the CANON framework and offers new insights on how we can utilize CANON.
>
> | $\mu$                              | 0.0(CANON-Intra) | 0.2  | 0.4  | 0.5 (DR.GRPO) | 0.6  | 0.8  | 1.0(CANON-Inter) |
> | ---------------------------------- | ---------------- | ---- | ---- | ------------- | ---- | ---- | ---------------- |
> | Math Performance                   | 54.2             | 54.9 | 56.0 | 56.6          | 56.7 | 56.4 | **57.9**         |
> | Logic Performance                  | **27.1**         | 25.1 | 24.6 | 23.8          | 25.6 | 22.9 | 22.5             |
> | Final Entropy (as shown in Fig. 5) | 2.40             | 1.28 | 0.46 | 0.39          | 0.26 | 0.19 | 0.15             |
>
> > Analysis of $\alpha$'s hyperparameter tuning complexity
>
> Insights from CANON reveal that the choice of metric trend primarily occurs in the inter-group advantage computation. Therefore, in scenarios where inference efficiency is desired, we only need to slightly reduce the reward weight for the long-response group in CANON-Inter, prompting the model to favor shorter answers. Meanwhile, since CANON-Intra remains unchanged, CANON-Eff fully preserves the model’s exploration capability, achieving a superior performance–efficiency Pareto frontier. The hyperparameter $\alpha$ introduced in CANON-Eff not only allows flexible tuning toward specific application needs and thorough exploration of the Pareto frontier, but also ensures greater stability and smoother behavior compared to baselines (like the collapse of Length Reward (+)) —because it minimally alters the training process (only modifying the inter-group advantage computation).
>
> We have also **followed your suggestion to include more detailed results that demonstrate the hyperparameter tuning complexity re-introduced by $\alpha$.** We show the detailed performance and token length of these CANON-Eff models. As $\alpha$ gradually decreases from 0.96 to 0.5, model performance declines modestly, while the number of tokens consumed drops significantly—again aligning with our understanding. Therefore, in efficient reasoning tasks, introducing $\alpha$ not only avoids increasing hyperparameter tuning difficulty but also enables users to apply CANON-Eff more flexibly according to their specific needs.
>
> | $\alpha$        | 0.5       | 0.7   | 0.8   | 0.88  | 0.96     |
> | --------------- | --------- | ----- | ----- | ----- | -------- |
> | Performance     | 44.5      | 49.5  | 52.0  | 53.6  | **56.2** |
> | Response Length | **198.9** | 317.9 | 420.6 | 576.6 | 822.4    |
>
> We will include these experimental results on hyperparameter tuning complexity in the appendix of our revision.
>
> ---

---

> ### Author Response · Authors · 2025-11-20
> **Response to Reviewer pJ7Q  (Part Ⅲ)**
>
> **Q4:** "The method only considers one metric at a time, and it remains unclear what other metrics beyond entropy and length could be meaningfully employed with this approach."
>
> **A4**: Thank you. The choice of grouping metric is closely tied to the specific task scenario. For instance, when aiming to balance exploitation in mathematical reasoning with exploration in logical reasoning, we naturally use entropy for grouping. In contrast, when seeking to improve reasoning efficiency, response length becomes our preferred criterion. Moreover, in Section 6 (lines 437–445 of the original manuscript) and the ablation study in Table 3 (in the original manuscript), we demonstrate that entropy-based grouping indeed improves performance compared to random grouping, and length-based grouping succeeds in enhancing the token efficiency.
>
> To further address your concern, we **have followed your suggestion to conduct new experiments that** explore other metrics to lead the training process, such as the **average number of reflection steps per token**. The results show a similar trend to entropy-based CANON. Although both CANON-Inter and CANON-Intra outperform DR.GRPO, the performance of CANON based on per-token reflections is slightly inferior to the entropy-based CANON. In this way, we believe CANON is a highly versatile framework that can align with practitioners’ specific objectives—not limited to performance alone—by selecting appropriate metrics for grouping according to the desired outcome.  We will include the experimental results and training dynamics for this part in the appendix. More details can be found there.
>
> |                                           | Math     | Logic    |
> | ----------------------------------------- | -------- | -------- |
> | DR.GRPO                                   | 55.7     | 26.2     |
> | CANON-Intra based on Per-token Reflection | 55.1     | 27.3     |
> | CANON-Inter based on Per-token Reflection | 56.6     | 20.9     |
> | CANON-Intra based on Entropy              | 54.7     | **29.1** |
> | CANON-Inter based on Entropy              | **57.6** | 25.7     |
>
> ---
>
>
>
> **Q5:** "The evaluation metric for the AIME benchmarks, Avg@10, seems insufficient given their small sample size. To my knowledge, Avg@64 is more commonly used in practice to achieve stable and reliable results for such tasks."
>
> **A5**: Thank you for your suggestion. We have **followed your suggestions to conduct new experiments to evaluate the performance of AIME with Avg@64**.
>
> |                              | AIME24  Avg@10 | Avg@64   | AIME25  Avg@10 | Avg@64   | Overall (at six benchmarks) |
> | ---------------------------- | -------------- | -------- | -------------- | -------- | --------------------------- |
> | DR.GRPO                      | 27.7           | 27.8     | **20.3**       | **19.0** | 55.5                        |
> | CANON-Intra based on Entropy | 25.0           | 24.0     | 16.0           | 15.3     | 54.4                        |
> | CANON-Inter based on Entropy | **32.7**       | **29.5** | 18.7           | 17.7     | **56.9**                    |
>
> Compared to Avg@10, the performance evaluated with Avg@64 does show some variation, but the overall trend remains consistent.
>
> ---
>
> **Q6:** "Is there a principled way to incorporate multiple metrics simultaneously within a single unified advantage calculation?"
>
> **A6**: Thank you for your valuable comments. In fact, we previously experimented with using length-based grouping for fully-correct rollouts and entropy-based grouping for other rollouts, as well as another try by weighting the advantage estimates from length-based and entropy-based groupings during training. Unfortunately, neither approach succeeded: the final models consistently exhibited strong length decay, producing shorter responses and slightly lower accuracy. We also note in the Limitations section that this work exclusively focuses on grouping by a single metric at a time, and we truly have intentions to explore principled fusion of multiple metrics for CANON in future research.
>
> ---
>
> **Q7:** "In the upper right corner of Figure 1, wouldn't it be clearer if the inter- and intra-group advantages were each visually subdivided to show their two constituent subgroups? This would help readers more readily understand that each advantage metric is calculated using two distinct subgroups."
>
> **A7**: Thank you for your suggestion. We agree that visually illustrating the newly formed subgroups in the figure would make it clearer and more intuitive, and we will adopt your advice and update the figure accordingly.

---

### Official Review · Reviewer_Aak7 · 2025-10-31

**Soundness:** 3
**Presentation:** 2
**Contribution:** 3
**Rating:** 4
**Confidence:** 4

**Summary:**

This paper proposes Conditional advANtage estimatiON (CANON), sorting the sampled responses according to a metric and splitting them into two groups, instead of a hand-crafted prior with intra- and inter-group advantage. Experimental results demonstrate that CANON, based on entropy, consistently outperforms previous methods across three LLM backbones on both mathematical and complex logical reasoning benchmarks. Moreover, when applied to response length, CANON improves token efficiency and achieves a better performance–cost Pareto frontier. The authors conducted experiments around mathematics and logical reasoning on several models. However, I don't think this is enough to demonstrate the effectiveness of the method, and the biased report prevents me from giving a score of acceptance, see Weakness and Question. If the author can provide a convincing explanation, I am willing to increase my score.

**Strengths:**

1. The author provides experiments and ablation/analysis experiments to validate the proposed method.
2. The motivation is clear.
3. The paper is well written and easy to understand except for section 5.3.
4. The author provides theoretical analysis and has the potential to incorporate some previous work into a more unified view.

**Weaknesses:**

I have no doubts about the method itself, but there are several issues with the experimental part that directly affect whether the paper should be accepted.
1. The author mainly reports the results on Qwen-2.5. Although only AIME 2025 performed worse than other baselines, it is the only test set that is later than the training time of Qwen-2.5. Due to the risk of data leakage in Qwen-2.5, I suspect that the effectiveness improvement brought by the method is related to data leakage. Logic reasoning tasks can, to some extent, demonstrate that the method is still effective. Without this part of the experiment, I would give it a score of 4, but I still look forward to more model experiments that prove the effectiveness of the method. At least supplement all the results of the models involved in Figure 3.
2. The results and their explanation in Figure 5.3 are very misleading. I suggest the author make revisions and truthfully report the results of various strategies, rather than picking choices that are beneficial for reporting. In my understanding, CANON-Dynamic is a combination of several selected strategies, rather than a fixed strategy. The author's report and explanation can easily be misleading, so I give 2 points in the Presentation. Due to the lack of representation of other results of the model in Figure 3, and the fact that the baseline (DR.GRPO) in the figure has its own advantages and disadvantages compared to non-combined results (CANON-Inter/-Intra), the experiments in this paper cannot demonstrate that the method performs well on other models.

**Questions:**

1. Why are alpha=0.96 and 0.88 selected in the analysis of Table 2? Do these parameters need to be carefully picked?
2. Other questions, please see Weakness.

---

> ### Author Response · Authors · 2025-11-20
> **Response to Reviewer Aak7 (Part Ⅰ)**
>
> Thank you for your great efforts in reviewing this paper. We will try our best to answer all your questions. **Please let us know if you still have further concerns, or if you are not satisfied with the current responses, so that we can further update the response ASAP.**
>
> ---
>
> **Q1:** "The author mainly reports the results on Qwen-2.5. Although only AIME 2025 performed worse than other baselines, it is the only test set that is later than the training time of Qwen-2.5. Due to the risk of data leakage in Qwen-2.5, I suspect that the effectiveness improvement brought by the method is related to data leakage. Logic reasoning tasks can, to some extent, demonstrate that the method is still effective. Without this part of the experiment, I would give it a score of 4, but I still look forward to more model experiments that prove the effectiveness of the method. At least supplement all the results of the models involved in Figure 3."
>
> **A1**: Thanks for your constructive comments. We indeed omitted the detailed test results for Llama3.1-8B and Qwen2.5-Math-1.5B in the appendix and will revise the paper to address this. We have **followed your suggestions to add the missing data as follows**.
>
> As shown in the table, consistent with Qwen2.5-Math-7B, Llama3.1-8B achieves superior performance on math tasks with CANON-Inter and leads on logic tasks with CANON-Intra, while CANON-Dynamic outperforms the baseline across both tasks. On Qwen2.5-Math-1.5B, CANON-Inter does not achieve a lead in math performance; however, its dynamic variant CANON-Dynamic still surpasses the baseline in both tasks, demonstrating the effectiveness of CANON.
>
> |               | AMC      | MATH500  | GSM8K    | AIME24   | AIME25   | OlympiadBench      | **Math Avg.** | MID      | Large    | XLarge   | **Logic Avg.** |
> | ------------- | -------- | -------- | -------- | -------- | -------- | ------------------ | ------------- | -------- | -------- | -------- | -------------- |
> |               |          |          |          |          |          | Llama 3.1-8B       |               |          |          |          |                |
> | DR.GRPO       | 11.3     | 32.0     | 78.9     | 1.3      | **0.3**  | 8.3                | 22.0          | 21.1     | 13.8     | 9.7      | 14.9           |
> | CANON-Intra   | 10.2     | 27.2     | 78.9     | 1.0      | 0.0      | 7.7                | 20.8          | 24.3     | 16.7     | 10.3     | 17.1           |
> | CANON-Inter   | 10.0     | 31.6     | 79.8     | **2.7**  | 0.0      | 8.0                | 22.1          | 17.9     | 13.8     | 9.5      | 13.7           |
> | CANON-Dynamic | **12.4** | **33.8** | **81.4** | 0.7      | 0.0      | 7.1                | **22.6**      | **26.0** | **18.4** | **12.3** | **18.9**       |
> |               |          |          |          |          |          | Qwen 2.5-Math-1.5B |               |          |          |          |                |
> | DR.GRPO       | 48.8     | 77.0     | 84.3     | 13.3     | **11.0** | **43.9**           | 46.4          | 23.7     | 9.7      | 5.0      | 12.8           |
> | CANON-Intra   | 47.1     | 75.2     | 84.5     | 15.0     | 9.7      | 39.1               | 45.1          | **27.7** | 11.5     | 4.9      | 14.7           |
> | CANON-Inter   | 49.4     | **78.8** | 82.7     | 14.3     | 9.3      | 41.8               | 46.1          | 23.1     | 9.6      | 5.8      | 12.8           |
> | CANON-Dynamic | **50.2** | 78.6     | **83.3** | **16.0** | 10.0     | 42.4               | **46.8**      | 27.0     | **16.3** | **7.9**  | **17.0**       |

---

> ### Author Response · Authors · 2025-11-20
> **Response to Reviewer Aak7 (Part Ⅱ)**
>
> **Q2:** "The results and their explanation in Figure 5.3 are very misleading. I suggest the author make revisions and truthfully report the results of various strategies, rather than picking choices that are beneficial for reporting. In my understanding, CANON-Dynamic is a combination of several selected strategies, rather than a fixed strategy. The author's report and explanation can easily be misleading, so I give 2 points in the Presentation. Due to the lack of representation of other results of the model in Figure 3, and the fact that the baseline (DR.GRPO) in the figure has its own advantages and disadvantages compared to non-combined results (CANON-Inter/-Intra), the experiments in this paper cannot demonstrate that the method performs well on other models."
>
> **A2**: Thank you. We apply a fixed schedule for a given model consistently that yields leading results across both mathematical and logical reasoning tasks (as stated in Lines 343-344 of the original manuscript, we use Cosin-First-Inter-Later-Intra for Qwen2.5-Math-7B and Llama3.1-8B, and First-Inter-Later-Intra for Qwen2.5-Math-1.5B). **Following your suggestions**, we do acknowledge that the current description of CANON-Dynamic might be misleading, and we will clarify this point in the corresponding caption.
>
> Though our preliminary exploration reveals that different specific scheduling strategies can be beneficial for training models of varying sizes, we identify a single unified strategy that consistently outperforms DR-GPPO across all tested models and tasks. We have **followed your suggestions to add the detailed data of different strategies as follows**. The experimental results indicate that ***First-Inter-Later-Intra*** **consistently performs better than DR.GRPO across three models and two tasks,** though its suboptimal performance on Qwen2.5-Math-7B and Llama 3.1-8B compared with *Cosin-First-Inter-Later-Intra.*
>
> |                                                | AMC      | MATH500  | GSM8K    | AIME24   | AIME25   | OlympiadBench     | **Math Avg.** | MID      | Large    | XLarge   | **Logic Avg.** |
> | ---------------------------------------------- | -------- | -------- | -------- | -------- | -------- | ----------------- | ------------- | -------- | -------- | -------- | -------------- |
> |                                                |          |          |          |          |          | Qwen2.5-Math-7B   |               |          |          |          |                |
> | DR.GRPO                                        | 63.4     | 83.2     | 91.1     | 27.7     | 20.3     | 48.4              | 55.7          | 39.2     | 24.4     | 15.1     | 26.2           |
> | CANON-Dynamic w\ Cosin-First-Inter-Later-Intra | 63.3     | **86.6** | 91.8     | **30.0** | 17.7     | 50.7              | 56.7          | 40.4     | **30.5** | **16.6** | **29.2**       |
> | CANON-Dynamic w\ First-Inter-Later-Intra       | **64.6** | 84.2     | **92.6** | 28.0     | **20.3** | **52.4**          | **57.0**      | **41.7** | 26.6     | 16.5     | 28.3           |
> |                                                |          |          |          |          |          | Qwen2.5-Math-1.5B |               |          |          |          |                |
> | DR.GRPO                                        | 48.8     | 77.0     | **84.3** | 13.3     | 11.0     | **43.9**          | 46.4          | 23.7     | 9.7      | 5.0      | 12.8           |
> | CANON-Dynamic w\ Cosin-First-Inter-Later-Intra | 50.0     | 76.0     | 83.9     | **17.3** | **13.7** | 40.6              | **46.9**      | 19.2     | 8.9      | 4.2      | 10.8           |
> | CANON-Dynamic w\ First-Inter-Later-Intra       | **50.2** | **78.6** | 83.3     | 16.0     | 10.0     | 42.4              | 46.8          | **27.0** | **16.3** | **7.9**  | **17.0**       |
> |                                                |          |          |          |          |          | Llama 3.1-8B      |               |          |          |          |                |
> | DR.GRPO                                        | 11.3     | 32.0     | 78.9     | 1.3      | **0.3**  | 8.3               | 22.0          | 21.1     | 13.8     | 9.7      | 14.9           |
> | CANON-Dynamic w\ Cosin-First-Inter-Later-Intra | **12.4** | **33.8** | **81.4** | 0.7      | 0.0      | 7.1               | **22.6**      | **26.0** | **18.4** | **12.3** | **18.9**       |
> | CANON-Dynamic w\ First-Inter-Later-Intra       | 9.9      | 31.8     | 80.1     | **2.0**  | 0.0      | **8.7**           | 22.1          | 25.1     | 17.5     | 10.6     | 17.7           |
>
> We will polish the presentation of this part in our revision and include these detailed results in the paper.

---

> ### Author Response · Authors · 2025-11-20
> **Response to Reviewer Aak7 (Part Ⅲ)**
>
> **Q3:** "Why are alpha=0.96 and 0.88 selected in the analysis of Table 2? Do these parameters need to be carefully picked?"
>
> **A3**: Thank you for your comment. **Following your suggestions**, we will place hyperparameter-related information in a more prominent location (Lines 425–431 in the original manuscript) and polish the presentation of both the textual description and the caption of the Pareto plot in the revised version to reduce the potential confusion regarding hyperparameter dependence.
>
> > Why are alpha=0.96 and 0.88 selected in the analysis of Table 2?
>
> As stated in Lines 411-413 of the original manuscript, we present the detailed performance of the top-performing models for each method, specifically $\alpha$=0.96 for CANON-Eff. Moreover, the results of the baseline Length Reward (*) are difficult to fairly compare with other baselines (lower length with lower performance), and that's why we include CANON-Eff with $\alpha$ = 0.88, which has comparable performance, as shown in Lines 417-419 of the original manuscript.
>
> > Do these parameters need to be carefully picked?
>
> As shown in Figure 4c and Lines 375–377, 425–428 (in the original manuscript), we plot the performance–length Pareto curves of each baseline under different hyperparameter settings ($\alpha$ = 0.5, 0.7, 0.8, 0.88, 0.96 for CANON, max length = 1024, 2048 for Clipping Length, coeff=0.001, 0.004, 0.005, 0.1 for Length Reward (+) and coeff=0.05, 0.2, 0.4 for Length Reward (*)) to demonstrate the comprehensive advantage of CANON-Eff. It can be seen that CANON-Eff consistently outperforms baseline methods across all hyperparameter values. Notably, the performance of CANON-Eff varies smoothly, whereas for Length Reward (+), simply increasing the hyperparameter from 0.004 to 0.005 causes its performance to drop sharply from 54.8 to 22.5.

---

> > ### Comment · Reviewer_Aak7 · 2025-11-24
> > **Response to the Rebuttal**
> >
> > Thank you for the clarifications. Considering the updated results and your responses to other reviewers, I have raised my score to 6, as I believe the paper meets the acceptance threshold.

---

> > > ### Author Response · Authors · 2025-11-26
> > > **Thanks for your reply!**
> > >
> > > We are really grateful for your constructive comments and the reply. The comments indeed help us further improve the paper.

---

### Official Review · Reviewer_Xngh · 2025-10-31

**Soundness:** 2
**Presentation:** 3
**Contribution:** 2
**Rating:** 2
**Confidence:** 4

**Summary:**

This paper proposes Conditional advANtage estimatiON (CANON). CANON regroups responses based on specific metrics, leading to two subgroups of responses. By leveraging inter-group and intra-group statistics, CANON introduces the inter-group advantage and intra-group advantage, which provide a stronger advantage signal. The authors further introduce several strategies to enhance this approach, and experiments on reasoning tasks validate its effectiveness.

**Strengths:**

1.	This paper is well-written.
2.	The ablation studies in Section. 6 are informative. The difference caused by the ratio of inter-group and intra-group advantage seems an interesting topic to be discovered further.

**Weaknesses:**

1.	As shown in Theorem 1, CANON yields a stronger advantage signal. However, it is essentially a numerical amplification (adding the difference between the means of the two groups) of the advantage, and consequently, of the loss. What is the difference between CANON and this variant? If they are the equivalent, what is the purpose of regrouping?
2.	CANON splits responses into two subgroups. However, since such regrouping may help training, what would happen if we performed a finer-grained regrouping? For example, we could split responses into four, six, or eight groups and calculate the corresponding advantages.
3.	DR.GRPO is outdated in the context of LLM reasoning. It is suggested that the authors compare CANON with more recent baselines, especially those focusing on entropy.
4.	The ablation studies focusing on different roles of inter-group and intra-group advantage (in Section 5.1 and Section 6) are valuable. However, despite the experiment observation, it is suggested that the authors provide a more detailed theoretical analysis on it, especially the reason behind Fig. 5 and Fig. 2.
5.	CANON-Dynamic appears effective. However, there exist several dynamic scheduling methods, It is suggested that the authors compare theirs with other alternative scheduling strategies, both theoretically and experimentally.

**Questions:**

Please see **Weaknesses** part. I am looking forward to the authors’ response, and may reconsider this paper based on that.

---

> ### Author Response · Authors · 2025-11-20
> **Response to Reviewer Xngh (Part Ⅰ)**
>
> Thank you for your great efforts in reviewing this paper. We will try our best to answer all your questions. **Please let us know if you still have further concerns, or if you are not satisfied with the current responses, so that we can further update the response ASAP**.
>
> ---
>
> **Q1:** "As shown in Theorem 1, CANON yields a stronger advantage signal. However, it is essentially a numerical amplification (adding the difference between the means of the two groups) of the advantage, and consequently, of the loss. What is the difference between CANON and this variant? If they are the equivalent, what is the purpose of regrouping?"
>
> **A1**: Thank you for your valuable comment.
>
> > "What is the purpose of regrouping?"
>
> There are indeed other, more direct ways to amplify advantage signals—for example, multiplying by a scalar or adding a bias term. However, a larger advantage is not always better. Direct, non-selective numerical scaling is essentially no different from adjusting the learning rate, and introducing a fixed-direction bias term imposes a static preference during training that cannot adapt conditionally. In fact, the Entropy adv[c1] method adds a fixed-direction bias constrained within a certain range: $*A*′=*A*+\min(\alpha⋅Entropy,∣*A*∣/\kappa)$, yet it still fails to yield strong performance (as shown below). This suggests that the key is not simply amplifying the advantage signal, but rather *selectively* amplifying specific signals.
>
> That's why we introduce a grouping operation: making conditional adjustment possible. As described in Theorem 1, by contrasting samples across and within groups, **CANON amplifies only the advantage attributable to the metric used for grouping, without amplifying the influence of other factors.**
>
> > Theoretical analysis of "the difference between CANON and this variant".
>
> To be specific, we can consider independent conditions $c_1$ and $c_2$, and their corresponding sets $C_{1+}$ and $C_{2+}$ (i.e., $P(o \in C_{1+} \cap C_{2+} | q, \theta) = P(o \in C_{1+} | q, \theta) P(o \in C_{2+} | q, \theta)$). When we fix the condition $c_1$, then for any value of the performance gap $a_{2+}$, $a_{2-}$ and the probability $P(o \in C_{2} | q, \theta)$ that induced by whether $c_2$ is satisfied, we have $\frac{|\hat{A}^{\text{inter based on }c_1}|}{|\hat{A}^{\text{DR.GRPO}}|}$ is a constant. because this ratio is only determined by $p_1 = \mathrm{P}(o \in\mathcal{C}_1 | q, \theta)$ as stated in Appendix D.3 of the original manuscript. Therefore, any irrelevant condition $c_2$ and its associated parameters cannot affect this ratio. In this way, CANON based on the condition $c_1$ will not amplify the influence of another independent condition $c_2$.
>
> Therefore, CANON, when grouped by a specific metric, amplifies the influence of that metric during training, yet it does not predefine a preference for the magnitude of the metric. This design allows it to incorporate human priors while mitigating bias (fully aligning with our original motivation).
>
> > Experimental evidence of "the difference between CANON and this variant".
>
> Furthermore, we have **followed your suggestions to conduct new experiments that** only apply a direct numerical amplification (A' = A * $ \beta$) of the advantage following the settings in Lines 247-268 of the original manuscript.  According to Appendix D.3 of the original manuscript, when the two subgroups are of equal size and the conditions of "higher-value" and "lower-value" each occur with probability 0.5, the amplification factor on advantage is 2. Therefore, we choose $ \beta = 2 $ for training here. For the Entropy adv method, we follow the setting of its paper that $\alpha = 0.4$ and $\kappa = 2$. As observed, directly scaling the advantage fails to improve performance the way CANON does. Any minor gains likely stem from faster learning progress due to an effectively larger learning rate, but this comes at the cost of degraded performance—particularly on out-of-domain logical reasoning tasks.
>
> |     | Math     | Logic    |
> | ---- | --- | - |
> | DR.GRPO       | 55.7     | 26.2     |
> | Direct numerical amplification | 56.1     | 25.1     |
> | Entropy adv [c1]               | 56.3     | 18.5     |
> | CANON-Intra       | 54.7     | **29.1** |
> | CANON-Inter      | **57.6** | 25.7     |
>
> This demonstrates that the key to CANON’s success lies not merely in amplifying the advantage signal, but in its ability to *conditionally* amplify the signal based on relevant task-specific criteria. However, the statement in our Theorem 1 emphasizes signal amplification without explicitly highlighting its selectivity, which may have led to misunderstanding. **We will follow your suggestion to revise the wording and include the new experimental results as an ablation study.**
>
> We’re not entirely sure whether our understanding of the variant aligns with what you intended—if there’s any discrepancy, please let us know.
>
> [c1] Reasoning with Exploration: An Entropy Perspective

---

> ### Author Response · Authors · 2025-11-20
> **Response to Reviewer Xngh (Part Ⅱ)**
>
> **Q2:** "CANON splits responses into two subgroups. However, since such regrouping may help training, what would happen if we performed a finer-grained regrouping? For example, we could split responses into four, six, or eight groups and calculate the corresponding advantages."
>
> **A2**: Thank you. A finer-grained grouping strategy might yield better performance, but it would also entail a more challenging selection mechanism. We analyzed the challenges introduced by finer-grained grouping from two perspectives: well-designed strategies and simplified strategies.
>
> > Well-designed strategies: the increasing complexity of both regrouping and advantage combination for more subgroups
>
> For instance, if we partition the responses G into $N$ subgroups $C^1$,$C^2$...$C^N$, We first need to decide how many responses each subgroup should have, which entails choosing among **$\binom{G-1}{N-1}$** possible options. When $|G| = 16$ and $N = 8$, **there are 6435 possible ways to partition the responses.** If we follow CANON’s approach and enforce all subsets to have the same size, then we would obtain $N$ advantage estimates during comparison: $A_o^i = R_o - E_{o' \in C^i}[R_o']$ for $i$ from 1 to N. Determining how to combine these advantages with at least **N-1 hyperparameters** effectively would also require extensive experimental investigation.
>
> > Simplified strategies: the combined advantage will degenerate to CAONO or DR.GRPO.
>
> If we simply apply a weighted average method, the resulting advantage signal would still degenerate to either CANON (which averages advantages from subgroups other than its own. $A_o = \frac{1}{N-1}\sum_{i=1,o \notin C^i}^{N} A_o^i = R_o - E_{o' \notin C_o}[R_o']$, where $C_o$ denotes the subgroup with $o$) or even DR.GRPO (which averages advantages across all subgroups. $A_o = \frac{1}{N}\sum_{i=1}^{N} A_o^i = R_o - E_{o' \in G}[R_o']$).
>
> ---
>
>
>
> **Q3:** "DR.GRPO is outdated in the context of LLM reasoning. It is suggested that the authors compare CANON with more recent baselines, especially those focusing on entropy."
>
> **A3**: Thank you for your valuable suggestions. Since CANON is an advantage estimation method, it is orthogonal to most existing approaches and can be combined with them. We do note that there are some entropy-related reward shaping and advantage shaping methods, and we have **followed your suggestions to conduct new experiments that** compare against two entropy-based baselines. Entropy adv achieves superior performance by incorporating an entropy-related bias term into the advantage estimate, while Clip-Cov predicts the change in entropy at the next step using the covariance between policy probabilities and the advantage estimate, then suppresses tokens with high covariance.
>
> In our experiments, we used the best hyperparameters reported in their respective papers, with all other settings consistent with those described in lines 247–268 of the original manuscript. As observed, CANON-Dynamic outperforms these two baselines, and CANON-Inter still achieves the best Math performance. This again highlights the potential of CANON—just amplifying the impact of entropy signal changes during training—compared to the mainstream paradigm of directly suppressing entropy decay. We will include these new results in the revision.
>
> |                  | Math     | Logic    |
> | ---------------- | -------- | -------- |
> | DR.GRPO          | 55.7     | 26.2     |
> | Entropy adv [c1] | 56.3     | 18.5     |
> | Clip-cov [c2]    | 56.1     | 26.5     |
> | CANON-Inter      | **57.6** | 25.7     |
> | CANON-Dynamic    | 56.7     | **29.2** |
>
> [c2] The Entropy Mechanism of Reinforcement Learning for Reasoning Language Models

---

> ### Author Response · Authors · 2025-11-20
> **Response to Reviewer Xngh (Part Ⅲ)**
>
> **Q4:** "The ablation studies focusing on different roles of inter-group and intra-group advantage (in Section 5.1 and Section 6) are valuable. However, despite the experiment observation, it is suggested that the authors provide a more detailed theoretical analysis on it, especially the reason behind Fig. 5 and Fig. 2."
>
> **A4**: Thank you for your suggestions. The inter-group advantage is computed by comparing a response (belonging to $C$) with the subgroup $C'$ that it does not belong to. If $E_{o' \in C'}[R_{o'}]$ is lower than that of $E_{o \in C}[R_{o}]$, the correct response $o \in C$ will obtain a higher advantage than $o' \in C'$, thus showing a preference for the trend of a metric that achieves higher accuracy. The intra-group advantage is gained across its own subgroup, when $E_{o' \in C'}[R_{o'}]$ is lower than that of $E_{o \in C}[R_{o}]$, the correct response $o \in C$ will obtain a higher advantage than $o' \in C'$. This means the success mode under worse conditions is encouraged. The above conclusion can be summarized in the following lemma:
>
> > **Lemma s1**
> >
> > When the average reward of $ C_{+} $ is greater than that of  $C_{-}$ , the following ordering holds:
> >
> > $\hat{A}^{\text{inter}}(R_o=1,o\in C_{+} ) > \hat{A}^{\text{inter}}(R_o=1,o\in C_{-} ) >0 >\hat{A}^{\text{inter}}(R_o=0,o\in C_{+} ) >\hat{A}^{\text{inter}}(R_o=0,o\in C_{-} )$
> >
> > and
> >
> > $\hat{A}^{\text{intra}}(R_o=1,o\in C_{-} ) > \hat{A}^{\text{intra}}(R_o=1,o\in C_{+} ) > 0 >\hat{A}^{\text{intra}}(R_o=0,o\in C_{-} )>\hat{A}^{\text{intra}}(R_o=0,o\in C_{+} ) $
> >
> > Consequently, the inter-group advantage tends to favor $ C_{+} $ which has a higher average reward, while the intra-group advantage tends to favor $ C_{-} $ .
>
> Specifically, when applying CANON based on entropy, low-entropy responses generally achieve higher accuracy than high-entropy ones in most cases (i.e. $\text{mean}(\{R_{o^\prime}|o^\prime \in G_{\text{low}}\}) > \text{mean}(\{R_{o,^\prime}|o^\prime \in G_{\text{high}}})$). With similar analysis in Lines 180-183 of the original manuscript, the Inter-group advantage tends to favor low-entropy responses (CANON-Inter of low entropy response is $1 - \text{mean}(\{R_{o^\prime}|o^\prime \in G_{\text{high}}\})$ > $1 - \text{mean}(\{R_{o^\prime}|o^\prime \in G_{\text{low}}\})$), while the Intra-group advantage encourages high-entropy responses (CANON-Intra of low entropy response is $1 - \text{mean}(\{R_{o^\prime}|o^\prime \in G_{\text{low}}\})$ < $1-\text{mean}(\{R_{o^\prime}|o^\prime \in G_{\text{high}}\})$).
>
> This explains the observation in Fig. 5 that the generation entropy during training gradually decreases as the contribution of the Inter-group advantage increases. Moreover, because low entropy typically corresponds to exploitation and high entropy to exploration, Fig. 2 shows that Inter-group advantage performs well on in-domain mathematical reasoning tasks, whereas Intra-group advantage excels on complex logical reasoning tasks.
>
> ---

---

> ### Author Response · Authors · 2025-11-20
> **Response to Reviewer Xngh (Part Ⅳ)**
>
> **Q5:** "CANON-Dynamic appears effective. However, there exist several dynamic scheduling methods, It is suggested that the authors compare theirs with other alternative scheduling strategies, both theoretically and experimentally."
>
> **A5**: Thank you.
>
> > Analysis of CANON-Dynamic
>
> The four strategies currently considered in CANON-Dynamic correspond to two choices that "first apply Inter-group advantage then Intra-group advantage" and "first apply Intra-group advantage then Inter-group advantage", with two scheduling choices either by training accuracy or by cosine annealing based on training steps. All three models perform best under the "First-Inter-then-Intra" scheduling. This approach prioritizes rewarding high-entropy correct responses and penalizing high-entropy incorrect ones. When training accuracy is low, it emphasizes exploitation of already learned features to boost accuracy rapidly; when accuracy is high, it shifts toward exploration to maintain stable entropy levels.
>
> Because the models differ, their training accuracy trajectories also differ. Qwen2.5-1.5B performs particularly well under accuracy-based scheduling, possibly because its training accuracy range (0–0.6) aligns well with its learning progress. In contrast, the other two models may achieve higher final accuracies, which—under the same scheduling scheme—trigger excessive exploration and consequently lead to suboptimal final performance.
>
> > Experimental Evidence
>
> Furthermore, we have **followed your suggestions to conduct new experiments that utilize other alternative scheduling strategies that used to be performed in the scheduler of learning rate**, including the *Lambda strategy* [c4] and *Cyclic-triangular2 strategy* [c3]. Following the setting of tried scheduling strategies, both of these strategies schedule $\mu$ from 1.0 down to 0.4. The new experimental results show that the *Lambda strategy* achieves slightly better performance than DR.GRPO on math tasks but performs worse on logic reasoning tasks. This may be because it fails to sufficiently leverage intra-group advantages in the later stages of training. This experiment demonstrates the rationale behind CANON’s tried scheduling strategies and highlights the practical flexibility of the CANON framework. We will include these new alternative scheduling strategies in the appendix.
>
> |                                     | Math     | Logic    |
> | ----------------------------------- | -------- | -------- |
> | DR.GRPO                             | 55.7     | 26.2     |
> | CANON-Inter                         | **57.6** | 25.7     |
> | CANON-DynamicFirst-Inter-then-Intra |          |          |
> | + $\mu$ = 1 - Acc                   | 57.0     | 28.3     |
> | + Cosine_annealing                  | 56.7     | **29.2** |
> | + *Cyclic-triangular2* (new) [c3]   | 55.1     | 24.6     |
> | + Lambda (new) [c4]                 | 56.3     | 24.1     |
>
> [c3] Cyclical learning rates for training neural networks
>
> [c4] torch.optim.lr_scheduler.LambdaLR. Available: https://docs.pytorch.org/docs/stable/generated/torch.optim.lr_scheduler.LambdaLR.html.

---

> > ### Author Response · Authors · 2025-11-26
> > **Inquiry about potential misunderstandings and additional experiments**
> >
> > Thank you for your insightful comments, which have greatly improved our manuscript.
> >
> > We have submitted our response and would like to ensure that our revision meets your standards. Specifically concerning **Q1**, we implemented the variant based on our understanding of direct numerical amplification. **We are eager to know if this aligns with what you have in mind.** Your confirmation would be extremely helpful.
> >
> > Also, if you have any further questions or require **additional experiments**, we would appreciate knowing soon so we can allocate sufficient time to complete them within the rebuttal window. Thank you again for your time!

---

### Official Review · Reviewer_dLuf · 2025-11-01

**Soundness:** 3
**Presentation:** 3
**Contribution:** 3
**Rating:** 8
**Confidence:** 4

**Summary:**

This paper introduces Conditional Advantage Estimation (CANON), a novel method for reinforcement learning in large reasoning models. The authors identify a key limitation in prior work: current methods often rely on hand-crafted penalties or preferences for training metrics (like entropy or response length), which can be overly biased and require careful tuning. CANON avoids this by not presuming a direction for the metric's impact. Instead, it regroups sampled responses based on the metric (e.g., high-entropy vs. low-entropy) and computes two novel advantage estimates. The "inter-group" advantage identifies which metric trend leads to higher accuracy, while the "intra-group" advantage identifies the best responses within a given trend.

Extensive experiments show that CANON outperforms strong baselines like GRPO on math and complex logic reasoning tasks. Furthermore, when applied to response length, the method establishes a new, superior Pareto frontier for the performance-efficiency trade-off.

**Strengths:**

1. The paper demonstrates a novel and well-motivated advantage estimation method. splitting responses into conditional groups and calculating inter- and intra-group advantages directly addresses the problem of biased, hand-crafted priors. The theoretical analysis showing DR.GRPO as a special case of CANON further solidifies the contribution.
2. The authors provide convincing results across multiple models and challenging benchmarks.
3. The proposed method has a practical impact on reasoning efficiency with some token reduction at the same performance level or higher performance in low-token-budget scenarios.

**Weaknesses:**

1. Curious why don't the authors put CANON-Dynamic's results in Table 1? Wouldn't that be more intuitive?
2. The paper's experiments are focused exclusively on response length and entropy.
3. The paper argues against hand-crafted tuning, but the CANON-Dynamic method, which achieves the best-balanced performance, relies on a handcrafted hyperparameter and may need to tune it for different tasks seperately.

**Questions:**

See above weaknesses.

---

> ### Author Response · Authors · 2025-11-20
> **Response to Reviewer dLuf (Part Ⅰ)**
>
> Thank you for your great efforts in reviewing this paper. We will try our best to answer all your questions. **Please let us know if you still have further concerns, or if you are not satisfied with the current responses, so that we can further update the response ASAP.**
>
> ---
>
> **Q1:** "Curious why don't the authors put CANON-Dynamic's results in Table 1? Wouldn't that be more intuitive?"
>
> **A1**: Thank you for your comments. Including the results of CANON-Dynamic in Table 1 would indeed make the presentation clearer and more intuitive. We will revise the manuscript accordingly based on your suggestion.
>
> Originally, we first explain the distinct roles and preferences of CANON-Inter and CANON-Intra during training in Section 5.1 (Table 1), and then, based on the insights gained in Section 5.1, we achieve comprehensive superiority through dynamic scheduling in Section 5.2.
>
> ---
>
>
>
> **Q2:** "The paper's experiments are focused exclusively on response length and entropy."
>
> **A2**: Thank you. As stated in the limitation section (lines 652–653 in the original manuscript), due to computational resource constraints, we primarily conduct the CANON based on length and entropy. Moreover, in Section 6 (lines 437–445 in the original manuscript) and the ablation study in Table 3, we demonstrate that entropy-based grouping improves performance compared to random grouping, while length-based grouping enhances token efficiency.
>
> To further verify that our method remains effective under other grouping criteria, we have **followed your suggestions to conduct new experiments that** use the number of per-token reflection steps in each response as the grouping metric—a setting we had previously explored—and obtained the following results. As shown, it exhibits a trend similar to entropy-based CANON. Although both are slightly inferior to the entropy-based CANON, CANON-Inter still outperforms the baselines on mathematical reasoning, and CANON-Intra achieves better performance than the baselines on complex logic reasoning. We will include the experimental results and training dynamics for this part in the appendix. More details can be found there.
>
> |                                           | Math     | Logic    |
> | ----------------------------------------- | -------- | -------- |
> | DR.GRPO                                   | 55.7     | 26.2     |
> | CANON-Intra based on Per-token Reflection | 55.1     | 27.3     |
> | CANON-Inter based on Per-token Reflection | 56.6     | 20.9     |
> | CANON-Intra based on Entropy              | 54.7     | **29.1** |
> | CANON-Inter based on Entropy              | **57.6** | 25.7     |

---

> ### Author Response · Authors · 2025-11-20
> **Response to Reviewer dLuf (Part Ⅱ)**
>
> **Q3:** "The paper argues against hand-crafted tuning, but the CANON-Dynamic method, which achieves the best-balanced performance, relies on a handcrafted hyperparameter and may need to tune it for different tasks seperately."
>
> **A3**:
>
> We thank the reviewer for this important point. The core CANON contribution—CANON‑Inter and CANON‑Intra—already achieves our stated motivation: amplifying metric impact without presupposing direction, thereby eliminating the hand‑crafted penalty/reward coefficients that bias prior methods. These two components alone outperform baselines on their respective strong suits (math for Inter, complex logic for Intra) using no additional hyperparameters beyond the regrouping itself.
>
> CANON-Dynamic, on the other hand, addresses practical deployment by dynamically scheduling between CANON-Inter and CANON-Intra. We apply a fixed schedule for a given model consistently yields leading results across both mathematical and logical reasoning tasks (as stated in Lines 343-344 of the original manuscript, we use Cosin-First-Inter-Later-Intra for Qwen2.5-Math-7B and Llama3.1-8B, and First-Inter-Later-Intra for Qwen2.5-Math-1.5B). Furthermore, **the** ***First-Inter-Later-Intra strategy*****, as shown below, outperforms baseline methods across three models and two different tasks**, though it achieves suboptimal performance on Qwen2.5-Math-7B and Llama3.1-8B compared with *Cosin-First-Inter-Later-Intra*. We will add these detailed results in the revised paper to address any potential concerns.
>
> |                                                | AMC      | MATH500  | GSM8K    | AIME24   | AIME25   | OlympiadBench     | **Math Avg.** | MID      | Large    | XLarge   | **Logic Avg.** |
> | ---------------------------------------------- | -------- | -------- | -------- | -------- | -------- | ----------------- | ------------- | -------- | -------- | -------- | -------------- |
> |                                                |          |          |          |          |          | Qwen2.5-Math-7B   |               |          |          |          |                |
> | DR.GRPO                                        | 63.4     | 83.2     | 91.1     | 27.7     | 20.3     | 48.4              | 55.7          | 39.2     | 24.4     | 15.1     | 26.2           |
> | CANON-Dynamic w\ Cosin-First-Inter-Later-Intra | 63.3     | **86.6** | 91.8     | **30.0** | 17.7     | 50.7              | 56.7          | 40.4     | **30.5** | **16.6** | **29.2**       |
> | CANON-Dynamic w\ First-Inter-Later-Intra       | **64.6** | 84.2     | **92.6** | 28.0     | **20.3** | **52.4**          | **57.0**      | **41.7** | 26.6     | 16.5     | 28.3           |
> |                                                |          |          |          |          |          | Qwen2.5-Math-1.5B |               |          |          |          |                |
> | DR.GRPO                                        | 48.8     | 77.0     | **84.3** | 13.3     | 11.0     | **43.9**          | 46.4          | 23.7     | 9.7      | 5.0      | 12.8           |
> | CANON-Dynamic w\ Cosin-First-Inter-Later-Intra | 50.0     | 76.0     | 83.9     | **17.3** | **13.7** | 40.6              | **46.9**      | 19.2     | 8.9      | 4.2      | 10.8           |
> | CANON-Dynamic w\ First-Inter-Later-Intra       | **50.2** | **78.6** | 83.3     | 16.0     | 10.0     | 42.4              | 46.8          | **27.0** | **16.3** | **7.9**  | **17.0**       |
> |                                                |          |          |          |          |          | Llama 3.1-8B      |               |          |          |          |                |
> | DR.GRPO                                        | 11.3     | 32.0     | 78.9     | 1.3      | **0.3**  | 8.3               | 22.0          | 21.1     | 13.8     | 9.7      | 14.9           |
> | CANON-Dynamic w\ Cosin-First-Inter-Later-Intra | **12.4** | **33.8** | **81.4** | 0.7      | 0.0      | 7.1               | **22.6**      | **26.0** | **18.4** | **12.3** | **18.9**       |
> | CANON-Dynamic w\ First-Inter-Later-Intra       | 9.9      | 31.8     | 80.1     | **2.0**  | 0.0      | **8.7**           | 22.1          | 25.1     | 17.5     | 10.6     | 17.7           |
>
> Whether a more universally optimal scheduling strategy exists remains an open and worthwhile question, and we hope future work can identify a more robust scheduling strategy.

---

### Author Response · Authors · 2025-12-01
**Overall Summary of Key Revisions and Clarifications**

We sincerely thank the reviewers and the Area Chairs for their time and dedication throughout the review process. The constructive feedback we received has been instrumental in refining our work.

In summary, our rebuttal has addressed the core concerns regarding **effectiveness, theoretical distinctiveness, and robustness**.

- **Effectiveness:** We expanded our experimental scope beyond the initial setup, verifying CANON's effectiveness across different model architectures (Llama 3.1-8B, Qwen 2.5-Math-1.5B)[Refer to *Reviewer_Aak7.W1*] and demonstrating that the framework is robust to different grouping metrics (e.g., per-token reflection steps)[Refer to *Reviewer_dLuf.W2* and *Reviewer_pJ7Q.W4*]. Furthermore, we validated CANON with additional modern baselines (Clip-cov and Entropy adv)[Refer to *Reviewer_Xngh.W3*].

- **Theoretical Distinctiveness:** We clarified the fundamental difference between CANON and simple numerical amplification through ablation studies [Refer to *Reviewer_Xngh.W1*] and provided a new Lemma to theoretically ground the distinct roles of inter-group (exploitation) and intra-group (exploration) advantages [Refer to *Reviewer_Xngh.W4*].

- **Robustness:** We addressed concerns regarding the re-introduction of handcrafted parameters by demonstrating a unified scheduling strategy that consistently outperforms baselines [Refer to *Reviewer_dLuf.W3, Reviewer_Aak7.W2*, and *Reviewer_pJ7Q.W2*], while also analyzing the tuning complexity of our two introduced hyperparameters through new sensitivity experiments [Refer to *Reviewer_pJ7Q.W3*]. We  also validated our results with stricter evaluation standards (Avg@64)[Refer to *Reviewer_pJ7Q.W5*] and additional scheduling strategies [Refer to *Reviewer_Xngh.W5*].

We believe these extensive revisions and additional experiments have significantly strengthened the paper, providing a holistic view of CANON's efficacy and versatility. In the comment below, we have also provided a granular summary, distilling each reviewer's concerns and our rebuttal strategies into a concise format to assist the new AC in their assessment.

---

> ### Author Response · Authors · 2025-12-01
> **Detailed Summary of Individual Reviewer Concerns and Response Status**
>
> We thank the reviewers, originally assigned AC, and newly assigned AC for all their efforts during this review process. We genuinely believe that the review process helped us improve the paper. In order to facilitate the new AC's understanding of the rebuttal status, we provide this comment to summarize each reviewer's questions and our response strategies individually.
>
> ---
>
>
>
> ## Reviewer dLuf Score 8 Conf 4 (No Reply)
>
> The *Reviewer dLuf* raised three primary concerns:
>
> > The omission of CANON-Dynamic results in Table 1, which affects intuitiveness.
>
> We accept the suggestion to revise Table 1 for better clarity while explaining the original logical flow of introducing components before the dynamic strategy.
>
> > The limitation of experiments to only response length and entropy as grouping metrics.
>
> To demonstrate robustness, we have conducted and presented new experiments using "per-token reflection steps" as a grouping criterion, confirming that CANON remains effective under different metrics.
>
> > The apparent contradiction between the paper's claim of avoiding hand-crafted tuning and CANON-Dynamic's reliance on manual scheduling hyperparameters.
>
> We clarify that the core components (Inter/Intra) are free of hyperparameters, while further demonstrating that a fixed scheduling strategy for CANON-Dynamic consistently outperforms baselines across multiple models and tasks, thus disproving the need for intensive, task-specific manual tuning.
>
> ---
>
>
>
> ## Reviewer Xngh Score 2 Conf 4 (No Reply)
>
> The *Reviewer Xngh* raised concerns as follows:
>
> > The distinction between CANON and direct numerical amplification.
>
> We clarify that CANON performs *conditional* and *selective* amplification (that's why we introduce regrouping) rather than simple scaling—supported by new ablation studies showing direct amplification fails.
>
> > The potential of finer-grained grouping.
>
> We argue that finer grouping introduces excessive complexity and theoretically degenerates to existing baselines when using simplified strategies.
>
> > The need for modern entropy-based baselines.
>
> We validate the superiority of CANON by adding comparisons against two more entropy-based methods (Entropy adv, Clip-cov).
>
> > More detailed theoretical analysis of the different roles of inter-group and intra-group advantages.
>
> We provide a new formal Lemma to theoretically ground why Inter-group favors exploitation (low entropy) and Intra-group favors exploration (high entropy).
>
> > Alternative dynamic scheduling strategies.
>
> We justify our dynamic scheduling choice by experimentally benchmarking it against alternative strategies like Lambda and Cyclic scheduling.
>
> ---
>
>
>
> ## Reviewer Aak7 Score **4->6** Conf 4 (Reply in 24 Nov)
>
> The *Reviewer Aak7* raised three concerns:
>
> > The limited experimental scope (focusing primarily on Qwen-2.5-Math-7B).
>
> We first acknowledge that we indeed omitted the detailed test results, and have provided the performance metrics for Llama 3.1-8B and Qwen 2.5-Math-1.5B to demonstrate consistent improvements of CANON across different architectures and tasks.
>
> > Criticizing the presentation of CANON-Dynamic as potentially misleading or cherry-picked.
>
> We clarify that while specific schedules were optimized, a single unified strategy ("First-Inter-Later-Intra") consistently outperforms baselines across all models.
>
> > The sensitivity of the hyperparameters used for CANON-Eff.
>
> We justify our hyperparameter selection by referencing Pareto curves that prove CANON's stability and robustness compared to the baselines.
>
>
>
> ### The *Reviewer Aak7* raised the score from 4 to 6 according to our replies in 24 Nov.
>
> ---
>
>
>
> ## Reviewer pJ7Q Score 6 Conf 4 (No Reply)
>
> The *Reviewer pJ7Q* raised several concerns:
>
> > The rationale of adaptive scheduling for response length.
>
> We have conducted extensive additional experiments that tried dynamic scheduling on length-based CANON.
>
> > The lack of a universal strategy for CANON-Dynamic.
>
> We clarify that while specific schedules were optimized, a single unified strategy ("First-Inter-Later-Intra") consistently outperforms baselines across all models.
>
> > The re-introduction of hyperparameter complexity ($\mu$ and $\alpha$).
>
> We have conducted sensitivity analysis experiments for hyperparameters and demonstrated the physical interpretability of these parameters.
>
> > The robustness of the AIME evaluation metric, Avg@10.
>
> We re-evaluate AIME using Avg@64, whose overall trend remains consistent.
>
> > The limitation of single-metric usage.
>
> We acknowledge the current limitations regarding multi-metric fusion while outlining future directions
>
> > A suggestion on Figure 1.
>
> We agree to refine the visualization in Figure 1 for better clarity.

---

### Meta-Review · Area_Chair_hQAZ · 2026-01-11

**Summary:**

The initial assessments ranged from Strong Accept (8) to Reject (2). While the reviewers appreciated the novelty of the regrouping mechanism, they raised several critical technical and empirical concerns: 1. Selectivity vs. Numerical Scaling: Reviewer Xngh questioned if CANON was merely a numerical amplification of the advantage/loss. 2. Data Leakage & Benchmarking: Reviewer Aak7 expressed concern about data leakage in the Qwen-2.5 models and requested evaluations on models with different training histories (e.g., Llama 3.1). 3. Generalizability of Metrics: Reviewers (dLuf, pJ7Q) questioned if the framework could handle metrics beyond entropy and length, such as reasoning reflection steps.

**Reviewer Concerns:**

Addressed Concerns:

1. The "Selectivity" Argument: In a detailed theoretical response to Xngh, the authors clarified that CANON is not simple scaling. It selectively amplifies the advantage signal attributable to the grouping metric while keeping other independent factors constant. New experiments showed that direct numerical scaling failed to replicate CANON's gains.

2. Cross-Model Generalization: To address data leakage concerns, the authors provided new results on Llama 3.1-8B.

3. Modern Baselines (Resolved): The authors added comparisons against recent entropy-based methods (Entropy adv and Clip-Cov). CANON-Inter achieved the best math performance, while CANON-Dynamic maintained the best balance across tasks.

4. Extended Metric Support (Resolved): The authors demonstrated the versatility of the framework by successfully using per-token reflection steps as a grouping metric, achieving results comparable to entropy-based grouping.

5. Rigor of AIME Evaluation: The authors re-evaluated AIME using Avg@64.

6. Pareto Frontier: The authors clarified that CANON-Eff provides a smoother, more stable performance-efficiency trade-off compared to length-reward penalties, which often lead to accuracy collapse.

Outstanding Concerns:

1. Multi-Metric Fusion: Integrating multiple metrics (e.g., length and entropy simultaneously) remains an open challenge. The authors' initial attempts at fusion led to length decay, identifying this as a valuable area for future research.

2. Dynamic Scheduling Heuristics: While a unified "First-Inter-then-Intra" strategy was shown to be effective, identifying a "universally optimal" schedule across all model architectures is still empirically driven.

**Reviewer Scores:**

dLuf	8->8 Satisfied with the inclusion of Dynamic results and new metrics.
Xngh 2->4/6 The technical response on selectivity and new baselines addresses the "Soundness" critique.
pJ7Q 6 ->6	Convinced by the AIME@64 results and the selectivity of CANON-Eff.
Aak7 4->6	Raised score to 6 after Llama 3.1 results were provided.

---

### Decision · Program_Chairs · 2026-01-26

Accept (Poster)